# Evolutionary Reasoning Does Not Arise in Standard Usage of Protein Language Models

**Yasha Ektefaie\* ‡**
Eric and Wendy Schmidt Center
Broad Institute of Harvard and MIT
Cambridge, MA 02142
yektefai@broadinstitute.org

**Andrew Shen\***
Department of Biomedical Data Science
Stanford University
Stanford, CA 94305
ashen7@stanford.edu

**Lavik Jain**
Department of Biomedical Informatics
Harvard Medical School
Boston, MA 02115
lavikjain@college.harvard.edu

**Maha Farhat**
Department of Biomedical Informatics
Harvard Medical School
Boston, MA 02115
maha_farhat@hms.harvard.edu

**Marinka Zitnik ‡**
Department of Biomedical Informatics
Harvard Medical School
Boston, MA 02115
marinka@hms.harvard.edu

## Abstract

Protein language models (PLMs) are often assumed to capture evolutionary information by training on large protein sequence datasets. Yet it remains unclear whether PLMs can reason about evolution—that is, infer evolutionary relationships between sequences. We test this capability by evaluating whether standard PLM usage, frozen or fine-tuned embeddings with distance-based comparison, supports evolutionary reasoning. Existing PLMs consistently fail to recover phylogenetic structure, despite strong performance on sequence-level tasks such as masked-token and contact prediction. We present PHYLA, a hybrid state-space and transformer model that jointly processes multiple sequences and is trained using a tree-based objective across 3,000 phylogenies spanning diverse protein families. PHYLA outperforms the next-best PLM by 9% on tree reconstruction and 23% on taxonomic clustering while remaining alignment- and guide-tree-free. Although classical alignment pipelines achieve higher absolute accuracy, PHYLA narrows the gap and achieves markedly lower end-to-end runtime. Applied to real data, PHYLA reconstructs biologically accurate clades in the tree of life and resolves genome-scale relationships among *Mycobacterium tuberculosis* isolates. These findings suggest that, under standard usage, evolutionary reasoning does not reliably emerge from large-scale sequence modeling. Instead, PHYLA shows that models trained with phylogenetic supervision can reason about evolution more effectively, offering a biologically grounded path toward evolutionary foundation models.

39th Conference on Neural Information Processing Systems (NeurIPS 2025).

# 1 Introduction

Protein language models (PLMs) use transformers with masked language or autoregressive self-supervision to model molecular sequences (Rives et al., 2021; Lin et al., 2022; Alley EC, 2019; Madani, 2023; Notin, 2022). PLMs have shown state-of-the-art performance across predictive (Meier et al., 2021; Rives et al., 2021; Rao et al., 2021; Elnaggar et al., 2021; Alley EC, 2019; Rao et al., 2020) and generative (Lin et al., 2022; Hayes et al., 2024; Madani, 2023; Ferruz, 2022) tasks. At their core, PLMs learn statistical distributions over amino acid sequences, modeling which residues are likely to appear given their surrounding context. This enables powerful zero-shot predictions of mutational effects by observing how the likelihood of a substitution changes under the model. Over time, this paradigm has been framed as a form of "evolutionary modeling," with the assumption that capturing residue-level distributions implicitly reflects evolutionary dynamics. In practice, researchers often quantify these relationships using distance measures—typically Euclidean or cosine distances—between frozen or fine-tuned embeddings Alley EC (2019); Hie et al. (2022); West-Roberts et al. (2024); Pantolini et al. (2024); Muir et al. (2025).

However, in this work, we argue that this paradigm conflates *evolutionary modeling* with *evolutionary reasoning*. Evolutionary modeling, as practiced by current PLMs, involves capturing local amino acid distributions to estimate residue likelihoods. In contrast, evolutionary reasoning requires a deeper level of abstraction: given a set of sequences, can a model infer the evolutionary relationships that generated them? This includes identifying shared ancestry, relative divergence, and tree-like structure — tasks that cannot be solved by distributional modeling alone.

This distinction is critical. Evolutionary reasoning is the foundation of phylogenetic analysis — a problem that predates modern biology and underlies centuries of effort to reconstruct how life has changed through time. Classical phylogenetics attempts to recover the tree of life Hug (2016) from observed sequences, but current PLMs are not evaluated on their ability to do this. They operate largely in isolation, modeling individual sequences without explicit multi-sequence comparison, structure discovery, or relational inference. Beyond phylogeny, important biological insights can be discerned from reasoning across sequences, whether it is determining the impact of a protein variant (Meier et al., 2021; Brandes, 2023; et al., 2023) or annotating functions of poorly characterized proteins (Nguyen et al., 2024; Avsec, 2021; Zvyagin et al., 2022; Queen et al., 2024).

This work poses the following question: *Are current PLMs capable of evolutionary reasoning? And if not, how must we reimagine their architecture and training to enable them to do so?*

**Present Work.** To address the limitations of current PLMs in performing evolutionary reasoning, we make several contributions. First, we introduce a benchmark for evolutionary reasoning over protein sequences, built from curated datasets containing ground-truth phylogenetic trees derived from established phylogenetic studies and databases. This benchmark enables rigorous evaluation of whether models can infer evolutionary structure from sets of sequences, rather than simply modeling local residue distributions. Second, we evaluate existing PLMs on this benchmark. Using their learned embeddings, we attempt to reconstruct phylogenetic trees and find that these models Lin et al. (2022); Hayes et al. (2024); ESM Team (2024); Nijkamp et al. (2022); Brixi et al. (2025) fail to outperform a simple baseline, Hamming distance, when evaluated under standard usage (frozen or fine-tuned embeddings with distance-based comparison). This analysis highlights a fundamental limitation: current PLMs are architecturally and functionally optimized for intra-sequence modeling, not inter-sequence reasoning.

To address this gap, we propose PHYLA, a new model architecture designed for evolutionary reasoning. PHYLA is a hybrid state-space and sparsified attention model that processes multiple sequences jointly. It alternates between inter-sequence modules, which detect conserved motifs across sequences, and intra-sequence modules, which contextualize these motifs within individual sequences—allowing the model to reason over sequence sets. We also introduce a novel pretraining objective tailored to phylogenetic structure. Instead of using masked language modeling, PHYLA is trained using a tree loss that teaches the model to embed sequences such that their pairwise distances reflect the correct evolutionary topology.

Despite its compact size (24M parameters), PHYLA outperforms existing PLMs in phylogenetic tree reconstruction and competitive accuracy in mutation effect prediction. On TreeBase—a dataset of protein sequences paired with phylogenetic trees from published studies—PHYLA outperforms much larger models, including ESM-3 (1.4B) and ProGen-XLarge (6.4B), by 8% and 13% respectively,

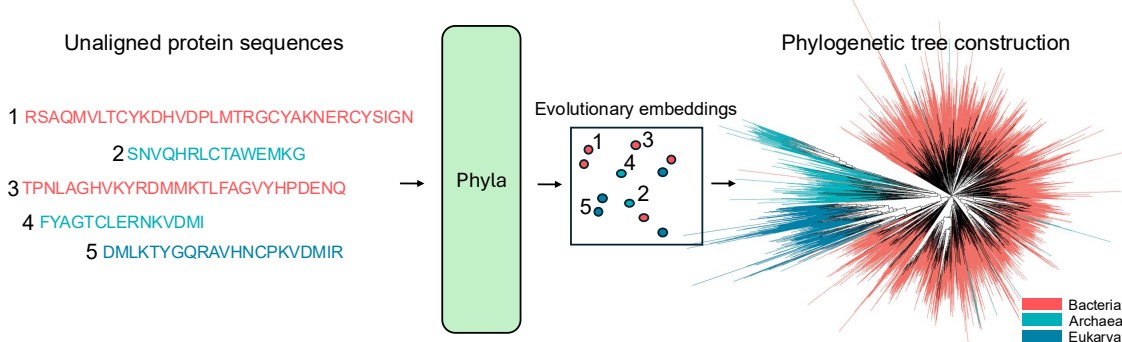

Figure 1: PHYLA takes unaligned protein sequences and generates embeddings that reflects relative evolutionary distance between input sequences which we can use to construct phylogenetic trees.

based on normalized Robinson-Foulds (normRF) distance. However, state-of-the-art accuracy remains with traditional phylogenetic algorithms Katoh et al. (2002); Price et al. (2010). PHYLA is substantially faster and narrows this gap, but further advances in model architecture and training objectives will be required to surpass classical methods. On the ProteinGym mutation effect prediction benchmark, PHYLA achieves a Spearman correlation of 0.64, matching or exceeding models with several orders of magnitude more parameters. We further demonstrate PHYLA's capacity for evolutionary reasoning by applying it to reconstruct the tree of life from ribosomal protein sequences spanning Bacteria, Archaea, and Eukarya. Additionally, we use PHYLA to infer whole-genome evolutionary relationships among *Mycobacterium tuberculosis* isolates. In both cases, the resulting trees deviate from canonical topologies and instead reflect known functional distinctions, suggesting that PHYLA learns representations aligned with true biological structure. These results position PHYLA as a foundation for a new generation of protein models grounded in evolutionary reasoning rather than sequence-level pattern recognition alone.

## 2    Related Work

**Protein Language Models (PLMs).** State-of-the-art protein language models include transformer-based models such as ESM2 (Lin et al., 2022), ESMC ESM Team (2024), ProGen (Madani, 2023), and Progen2 Nijkamp et al. (2022) that are trained using masked or autoregressive language modeling. These models learn to model the language of proteins by learning the co-occurrence of amino acid residues within a diverse training set. Other PLMs, such as ESM3 (Hayes et al., 2024), model additional data modalities. ESM3 considers structural and functional information in addition to the background amino acid sequences. These models have demonstrated good performance on intra-sequence reasoning from sequence modeling pre-training tasks, but have not explicitly been trained to perform inter-sequence reasoning between different sequences in the training set.

**Alternatives to self-attention.** Self-attention is the backbone of the transformer but suffers from quadratic scaling with sequence length, making modeling longer protein sequences difficult (Vaswani et al. (2017)). The Mamba state-space architecture has been proposed as an alternative backbone architecture for sequence-based foundation models. The architecture builds upon the S4 class of structured state-space models (Gu et al. (2022)) by adding a selection mechanism and a hardware-aware parallel algorithm. These advances allow Mamba to model long sequences efficiently. Beyond Mamba, other approaches use similar ideas to extend context length, including Hyena (Poli et al., 2023a) and xLSTM (Beck et al., 2024). More recent work has shown hybrid state-space and transformer architectures allow for long-context modeling while maintaining the advantages of transformers Poli et al. (2023b); Ren et al. (2025).

**Bioinformatics approaches to phylogenetic analysis.** Traditional tree reconstruction methods for a set of input protein sequences consist of generating a multiple sequence alignment (MSA) using one of many alignment algorithms. The MAFFT and Clustal Omega alignment algorithms are popular choices for efficient and accurate MSA generation (Katoh et al. (2002); Sievers et al. (2011)). These alignment algorithms align the input sequences by matching the location of the most conserved amino acids within the sequences. After generating the MSA, a phylogenetic tree is reconstructed using a

tree reconstruction algorithm, like FastTree and IQTree (Price et al. (2010); Nguyen et al. (2014)). These algorithms infer the structure of the phylogenetic tree with and without parametric models and usually with various heuristics to generate the most likely phylogenetic tree topology. The primary limitation of tree reconstruction is runtime inefficiency as tree sizes grow. More recent works can be found in the Appendix A.6.

# 3 Problem Definition

**Evolutionary Reasoning: Problem Formulation**    Let $S = \{s_1, \ldots, s_n\}$ denote a set of protein sequences drawn from an unknown evolutionary process that produced a true, strictly binary phylogenetic tree $T^*$ whose leaves correspond bijectively to $S$. The patristic (evolutionary) distance between sequences $s_i$ and $s_j$ on $T^*$ is denoted $d_{\mathrm{evol}}(i, j)$.

Our goal is to learn an encoder $f_\theta : \Sigma^* \to \mathbb{R}^d$ that maps each sequence to an embedding space, together with a distance function $d_\phi : \mathbb{R}^d \times \mathbb{R}^d \to \mathbb{R}_{\geq 0}$, such that the embedding-based distances

$$D_{ij} = d_\phi\big(f_\theta(s_i), f_\theta(s_j)\big)$$

approximate the true evolutionary distances $d_{\mathrm{evol}}(i, j)$.

A model $f_\theta$ is said to perform *evolutionary reasoning* on $S$ if there exists a computable mapping $h$ (e.g., a distance-based tree builder) such that the reconstructed tree

$$\hat{T} = h(D_{\mathrm{pred}}),$$

where $D_{\mathrm{pred}}[i, j] = D_{ij}$, minimizes a topology-level loss

$$\mathcal{L}_{\mathrm{tree}}(\hat{T}, T^*) = \mathrm{dist}(\hat{T}, T^*),$$

with $\mathrm{dist}$ denoting a structural metric such as normalized Robinson-Foulds (normRF) distance.

In our experiments, we use neighbor joining for $h$, Euclidean distance for $d_\phi$ and the normalized Robinson-Foulds distance for $\mathcal{L}_{\mathrm{tree}}$. Under this formulation, *evolutionary reasoning* corresponds to learning embeddings and distance functions whose geometry maximally preserves the information contained in the true phylogenetic structure.

**Alignment-Free and Guide-Tree-Free**    Unlike most prior methods, our approach does not rely on a multiple sequence alignment (MSAs are used solely to derive supervision distances, not as model inputs) or a predefined guide tree (i.e. a fixed topology to scaffold inference). Removing both introduces new challenges: alignment and tree inference are jointly NP-hard WANG & JIANG (1994); Roch (2005), and guide trees offer strong structural priors. By dispensing with both, PHYLA must discover meaningful patterns directly from unaligned sequences—capturing evolutionary structure without relying on handcrafted heuristics.

# 4 Methods

## 4.1 PHYLA Model Architecture

To do well on evolutionary reasoning, models must be able to perform good inter- and intra-sequence reasoning. **Inter-sequence reasoning** requires the model to detect motifs that recur across different sequences, revealing shared evolutionary signatures. **Intra-sequence reasoning** requires the model to contextualize identified motifs within the context of an individual sequence to determine evolutionary divergence from the identified motif.

To meet these requirements, we introduce a hybrid state-space transformer model, PHYLA. The PHYLA architecture alternates between inter-sequence and intra-sequence reasoning blocks. Inter-sequence reasoning is performed using BiMamba layers, a parameter-efficient state-space module with gated linear recurrence and deep receptive fields, enabling the model to detect long-range evolutionary motifs across sequences  Schiff et al. (2024). Intra-sequence reasoning is handled by sparsified attention layers to contextualize these motifs within each sequence. The current 24M-parameter PHYLA model comprises three inter-sequence blocks, each containing 16 BiMamba layers followed by a single sparsified attention layer. Each layer operates with a hidden dimension of 256.

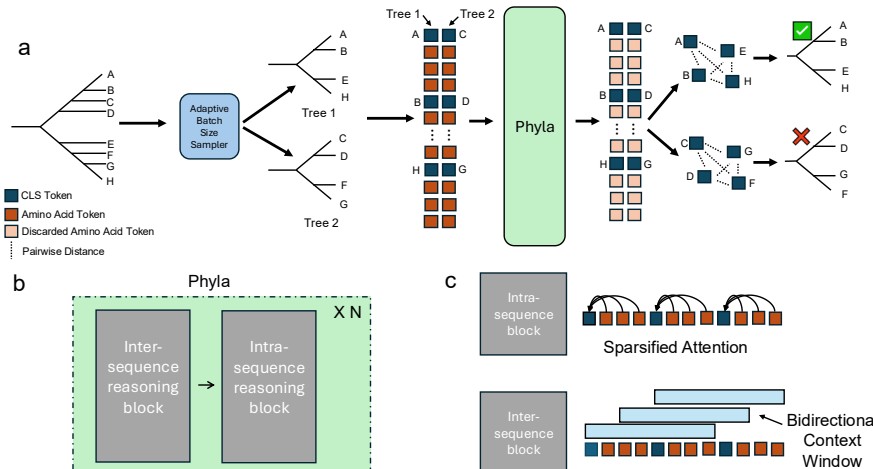

Figure 2: Overview of the PHYLA. (a) PHYLA processes batches of protein sequences sampled from a tree, learning embeddings used to reconstruct phylogenies. (b) It alternates inter- and intra-sequence reasoning blocks. (c) Inter-sequence reasoning uses long bidirectional convolutions to extract shared motifs; intra-sequence reasoning applies sparsified attention to maintain per-sequence context.

Unlike most PLMs which operate on a single sequence at a time (Lin et al. (2022); Hayes et al. (2024); ESM Team (2024); Notin et al. (2023b); Nijkamp et al. (2022)), **PHYLA is the first PLM designed to process multiple sequences concurrently**. This enables it to directly compare sequences, identify conserved motifs, and exploit these inter-sequence relationships to infer evolutionary structure at inference time. While prior models have employed hybrid state-space transformer architectures (Poli et al. (2023b)), PHYLA introduces a sparsification mechanism: an attention mask $M$ that constrains each token to attend only to tokens within its own sequence, preserving intra-sequence context while enabling efficient multi-sequence processing. Specifically $M$:

$$M_{ij} = \begin{cases} 1, & \text{if the } j\text{-th token is within the } i\text{-th sequence,} \\ 0, & \text{otherwise.} \end{cases} \tag{1}$$

## 4.2 PHYLA Model Training

**Tree Loss Pretraining.** To teach PHYLA how to model input sequences to recapitulate evolutionary relationships, we designed a novel tree loss pretraining function. Constructing a phylogenetic tree contains a series of decisions in how to split sequences based on their relative distance to each other. With this in mind, we use a *quartet loss* to supervise training. The quartet loss encourages the model to embed sequences such that inferred distances preserve correct quartet topologies. For a set of $n$ sequences with predicted pairwise distances $D_{\text{pred}} \in \mathbb{R}^{n \times n}$ and true distances $D_{\text{true}} \in \mathbb{R}^{n \times n}$, we sample a set of quartets $\mathcal{Q} = \{(i, j, k, \ell)\}$. For each quartet, we compute three possible pairwise distance sums:

$$x_{ij|k\ell} = D_{ij} + D_{k\ell},$$
$$x_{ik|j\ell} = D_{ik} + D_{j\ell},$$
$$x_{i\ell|jk} = D_{i\ell} + D_{jk},$$

and convert them into logits using a softmax-like transformation:

$$\text{logits}_q = \frac{1}{T^*} \left( \begin{bmatrix} x_{ij|k\ell} & x_{ik|j\ell} & x_{i\ell|jk} \end{bmatrix} - \text{mean}(x_{ij|k\ell}, x_{ik|j\ell}, x_{i\ell|jk}) \right),$$

where $T^*$ is a temperature hyperparameter controlling softness. We compute the cross-entropy loss between these logits and the ground-truth minimum sum (from $D_{\text{true}}$):

$$\mathcal{L}_{\text{quartet}} = \frac{1}{|\mathcal{Q}|} \sum_{q \in \mathcal{Q}} \text{CrossEntropy}(\text{logits}_q, \arg\min_{\alpha \in \{ij|k\ell, ik|j\ell, i\ell|jk\}} x_\alpha^{\text{true}}).$$

Quartet loss is the sole pretraining objective used to train PHYLA. **In contrast to most protein language models, which rely on masked language modeling (MLM) over amino acids, PHYLA does not use MLM.**

**Training Details.** PHYLA was trained on distances derived from 3,321 high-quality multiple sequence alignments (MSAs) curated from the OpenProteinSet Ahdritz et al. (2023). Low-quality MSAs were removed using the filtering procedure described in EVE Frazer et al. (2021). For each MSA, we compute pairwise distances between sequences by counting the number of matching positions and dividing by the total alignment length. These normalized pairwise distances are used as supervision during training. We used MSA-derived pairwise distances rather than tree-based distances, as experiments showed that training on tree distances did not yield significant performance gains (Appendix A.1.2). Using the precomputed MSAs from OpenProteinSet therefore provided comparable results at substantially lower computational cost. The current 24M parameter model was trained on a single 80GB H100 GPU for 3 days with the AdamW optimizer using a 10,000 step linear warmup up to a learning rate of 1e-5 (Loshchilov & Hutter (2019)).

**Adaptive Batch Size Sampling.** We employ an adaptive batch sizing approach to efficiently utilize GPU memory and avoid overfitting to a specific tree topology. We determine the largest subtree $t \in T$ at every training step that can fit within the available GPU memory. Next, we randomly sample a subtree size $n$ such that $10 \leq n \leq |t|$, where $|t|$ is the number of sequences in $t$. Finally, we identify how many subtrees of the sampled size $|t|$ can be accommodated within the GPU memory. If the model encounters an out-of-memory (OOM) error during this process, the subtrees are resampled with both the subtree size and the number of subtrees halved. We empirically determined that PHYLA can process input lengths up to 213,350 tokens on a 32 GB GPU and up to 302,350 tokens on a 48 GB GPU. For other GPU memory sizes, we used a linear model to estimate the maximum allowable input length. Given the length of the longest protein in the input, we computed the maximum number of sequences that could fit within the memory limit.

**Training Procedure.** During training, a phylogenetic tree $T$ is sampled, where $T$ consists of $N$ sequences $S$. Each sequence is tokenized with an alphabet of 23 tokens, corresponding to 20 standard amino acids, a CLS token, a mask token, and a pad token. The input to PHYLA is $S$ with a $[CLS]$ token concatenated in front of each tokenized sequence, $s \in S$: $\{[CLS]s_1\|[CLS]s_2\|[CLS]s_3...[CLS]s_n\}$. The size and number of trees considered in each training step are determined at each training step by the adaptive batch size sampler.

## 5 Experiments

**Datasets.** We evaluate PHYLA on a range of evolutionary and functional reasoning tasks. For phylogenetic tree reconstruction, we use two held-out datasets: TreeBase, which includes 1,533 curated phylogenetic trees across diverse species (Piel & Tannen (2009)), and TreeFam, which contains 9,586 gene-family trees spanning a wide evolutionary range (Li et al. (2006)). Dataset statistics are provided in Table 3 and Table 4, with further details in Appendix A.1.1. To assess taxonomic classification, we use bacterial isolate sequences from the Genome Taxonomy Database (GTDB) (Parks et al. (2021)). We define five classification tasks at different levels of taxonomic granularity: Class, Order, Family, Genus, and Species. These tasks measure the model's ability to cluster sequences according to hierarchical evolutionary relationships. Together, these seven tasks—tree reconstruction in TreeBase and TreeFam, and taxonomic classification at five levels using GTDB—form our evolutionary reasoning benchmark. To evaluate performance beyond evolutionary structure, we also assess performance on functional prediction using the ProteinGym benchmark, which consists of 83 protein mutation effect datasets (Notin et al., 2023a) (Appendix A.1.3).

**Baselines.** We consider four protein language models, one genomic foundation model, six models from the ProteinGym benchmark, and a naive Hamming distance baseline. The protein language models include ESM2, ESM3, ESM C, and ProGen2 (Lin et al. (2022); Hayes et al. (2024); ESM Team (2024); Nijkamp et al. (2022)). The genomic foundation model is Evo 2 (Brixi et al. (2025)). The six models from the ProteinGym benchmark are ProteinNPT, MSA Transformer, ESM-1v, Tranception, TranceptEVE, and DeepSequence (Notin et al. (2023a,b); Rao et al. (2021); Meier et al. (2021); Notin et al. (2022); Riesselman (2018); Notin (2022)). We also evaluate a traditional phylogenetic pipeline consisting of a multiple sequence alignment constructed via MAFFT Katoh et al. (2002) and then a tree constructed via FastTree Price et al. (2010).

**Evaluation setup.** We consider three evaluation settings. **Tree reconstruction**: This setting evaluates the model's ability to reconstruct phylogenetic trees given solely the original sequences. For each method, we generated protein embeddings, computed pairwise distances between embeddings, and reconstructed trees using the neighbor-joining algorithm. We evaluate tree reconstruction by comparing the predicted tree to the reference tree using the normalized Robinson-Foulds metric (Robinson & Foulds (1981)). **Taxonomic clustering**: This setting evaluates the model's ability to perform fine-grained and coarse-grained classification of sequences. We cluster 500 sequences at each taxonomic level of Class, Order, Family, Genus, and Species and assess the homogeneity of the predicted clusters. **Functional prediction**: This setting evaluates the model's ability to predict functional labels given solely the original sequences. We assess functional prediction by training a linear probe classifier on the generated embeddings, and also by utilizing the predicted tree structure to assign labels. More details can be found in Appendix A.1.1.

## 5.1 PHYLA can Reason over Protein Sequences

**Experimental setup.** To assess the ability of PHYLA to reason over sequences, we assess PHYLA's ability to reconstruct phylogenetic trees on the TreeBase and TreeFam datasets. We use the metric of Robinson-Foulds distance, or "RF", whereby a larger RF value is equivalent to a larger distance between predicted and reference tree, and can be interpreted as a lower quality predicted tree. The RF metric is not invariant to tree size, so we compute the normalized RF, or "normRF", to directly compare the tree reconstruction performance between trees of different sizes. We utilize the ETE3 Toolkit implementation of RF and normRF distance (Jaime Huerta-Cepas & Bork (2016)). In addition, we assess PHYLA's ability to cluster sequences across various taxonomic levels. We report the metric of homogeneity across predicted clusters as a measure of the ability to classify sequences. We compare the performance of PHYLA against PLMs (ESM2, ESM3, ESM C, ProGen2), a genomic foundation model (Evo 2), and a traditional phylogenetics workflow (MAFFT and FastTree) (Lin et al. (2022); Hayes et al. (2024); ESM Team (2024); Nijkamp et al. (2022); Brixi et al. (2025); Katoh et al. (2002); Price et al. (2010)). Table 1 shows the normRF performance of PHYLA and benchmark models on the TreeBase and TreeFam datasets.

**Results.** PHYLA achieves the best performance out of all tested PLMs on both the TreeBase and TreeFam tree reconstruction benchmarks, outperforming all tested baselines—including models with 12 to 266 times more parameters, such as ESM2 (650M) and ProGen2-XLarge (6.4B) (Table 1). On TreeFam, PHYLA reduces normRF by 13.4% compared to the next-best model (ESM2 650M). This improvement is not attributable to data overlap with the pretraining corpus (Appendix A.1.1). Fine-tuning ESM2 embeddings with the same tree-based loss did not yield meaningful gains (Appendix A.1.2), and training PHYLA with tree-derived versus MSA-derived supervision produced comparable results (Appendix A.1.2), indicating that performance arises from the model architecture rather than specific training heuristics.

Traditional alignment-based methods remain the accuracy upper bound: the MAFFT + FastTree pipeline achieves a normRF of 0.65 on TreeBase and 0.32 on TreeFam, while PHYLA attains 0.73 and 0.58, respectively—closing roughly half of this gap. Notably, the classical workflow required roughly 2 hours on TreeBase and 66 hours on TreeFam across multiple CPU nodes, whereas PHYLA completed both in under an hour on a single H100 GPU. On the taxonomy clustering benchmark, PHYLA also achieves the best performance across all taxonomic levels, improving species-level homogeneity by 23.2% over the strongest baseline. Notably, all PLMs underperform compared to the naive Hamming distance baseline on TreeBase, highlighting the challenge of this dataset. These results underscore the effectiveness of incorporating evolutionary structure into PHYLA architecture and training.

## 5.2 PHYLA Trees Encode Protein Functional Information

**Experimental setup.** To evaluate the expressivity of the learned embeddings, we use two complementary strategies for PHYLA and the baseline PLMs. For PHYLA, we convert its embeddings into pairwise distances and construct trees using the neighbor-joining algorithm, as in the previous benchmark. Functional labels from the training sequences are then overlaid onto the resulting tree. Local sequence clusters are identified using TreeCluster (Balaban M, 2019), which assigns each evaluation sequence to a cluster. The functional label of a test sequence is inferred by averaging the labels of training samples within its cluster, and performance is measured using Spearman rank correlation

Table 1: **Evolutionary reasoning benchmark comparing PHYLA to existing protein language models (PLMs).** Tree reconstruction performance on TreeBase and TreeFam is evaluated using normalized Robinson-Foulds distance (normRF), where lower values indicate better agreement with ground truth phylogenies. Clustering performance at the Class, Order, Family, Genus, and Species levels is measured using homogeneity scores, with higher values indicating more accurate unsupervised grouping according to known taxonomic labels. For other clustering metrics see Appendix A.5.

| Model | TreeBase ↓ | TreeFam ↓ | Class ↑ | Order ↑ | Family ↑ | Genus ↑ | Species ↑ |
|---|---|---|---|---|---|---|---|
| Hamming Distance | 0.75 | 0.75 | – | – | – | – | – |
| MAFFT+FastTree | **0.65** | **0.32** | – | – | – | – | – |
| ESM2 (650M) | 0.78 | 0.67 | 0.64 | 0.66 | 0.68 | 0.71 | 0.75 |
| ESM2 (3B) | 0.79 | 0.67 | 0.55 | 0.56 | 0.57 | 0.59 | 0.67 |
| ESM3 (1.4B) | 0.81 | 0.72 | 0.61 | 0.63 | 0.66 | 0.67 | 0.72 |
| ESM C (300M) | 0.77 | 0.71 | 0.57 | 0.60 | 0.62 | 0.67 | 0.71 |
| ESM C (600M) | 0.80 | 0.73 | 0.61 | 0.66 | 0.66 | 0.71 | 0.75 |
| Evo 2 (7B) | 0.84 | 0.84 | 0.50 | 0.54 | 0.55 | 0.55 | 0.64 |
| ProGen2-Large (2.7B) | 0.77 | 0.68 | 0.60 | 0.65 | 0.66 | 0.71 | 0.75 |
| ProGen2-XLarge (6.4B) | 0.86 | 0.82 | 0.52 | 0.55 | 0.57 | 0.61 | 0.68 |
| PHYLA (24M) | 0.73 | 0.58 | **0.71** | **0.76** | **0.87** | **0.93** | **0.98** |

between predicted and true labels. For baseline PLMs, we follow the standard evaluation protocol and train a linear probe to predict functional labels directly from their embeddings. Applying this linear-probe approach to PHYLA performed poorly. We utilize the 83 datasets from the ProteinGym (Notin et al. (2023a)) benchmark as our evaluation set (Appendix A.1.3). Figure 3 shows the average Spearman correlation metric with both the dataset size (in number of sequences trained on) and the model size (in parameters) on the 83 ProteinGym evaluation datasets.

**Results.** PHYLA achieves competitive performance on the ProteinGym benchmark, ranking among the top-performing models across the 83 functional prediction tasks (Figure 3). Despite being trained on 34 times less data and with 27 times fewer parameters than models like ESM2 (650M), PHYLA achieves a comparable Spearman correlation of 0.64. Notably, PHYLA outperforms several larger PLMs, including ESM3 (1.4B) and Tranception, demonstrating that its tree-based representations encode meaningful biological function. These results suggest that the structured embeddings produced by PHYLA not only capture phylogenetic and taxonomic information but also generalize to downstream functional prediction—despite minimal model size and data requirements.

## 5.3 Ablation Analyses

**Experimental setup.** To understand the effect of the tree loss versus traditional masked language modeling loss, we trained PHYLA with only masked language modeling (PHYLA-MLM). To probe the effect of PHYLA architecture on performance, we ablated the sparsified attention layers, which are the intra-sequence reasoning blocks (PHYLA-NoAttention). We then evaluated both ablated PHYLA models on the tree reasoning benchmark and on functional prediction.

**Results.** As shown in Table 2, pretraining on a masked language modeling (MLM) objective resulted in substantial performance degradation across all tasks: a 19.3% increase in normalized Robinson-Foulds (normRF) distance for tree reconstruction (TreeBase and TreeFam), a 16.4% drop in Spearman correlation for functional prediction (ProteinGym), and a 5.3% reduction in average taxonomic clustering homogeneity. Similarly, ablating sparsified attention led to even greater degradation: a 20.4% increase in normRF, a 45.5% drop in functional prediction, and a 7.4% reduction in taxonomic clustering. These results highlight that both the tree-based training objective and the sparsified attention mechanism are critical to PHYLA's overall performance.

## 5.4 PHYLA application to real world datasets

PHYLA demonstrates promising performance in sequence reasoning. To showcase its capabilities, we applied PHYLA to the task of phylogenetic tree construction. The tree of life is a fundamental

Figure 3: **Functional prediction performance on ProteinGym with model sizes and dataset sizes.** Spearman correlation of existing PLMs and PHYLA on ProteinGym benchmark as a function of pretraining dataset size. Model points are sized based on the number of parameters in the model.

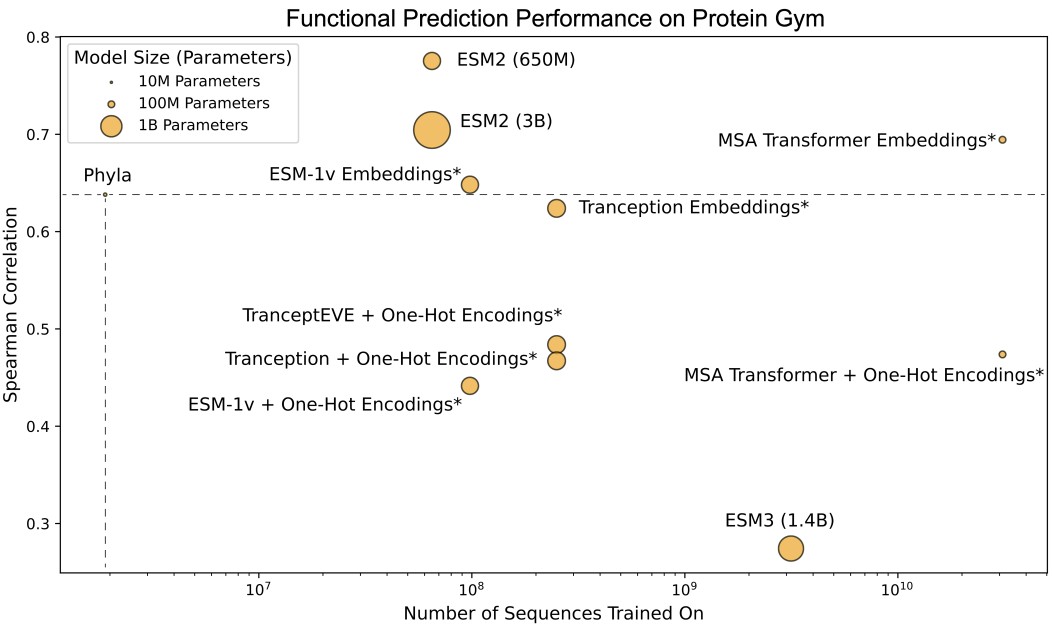

Table 2: **Ablation study on PHYLA's architecture and training objectives.** Tree reconstruction is evaluated on TreeBase and TreeFam using normalized Robinson-Foulds distance (normRF), where lower values indicate better alignment with ground truth trees. Functional prediction is assessed on ProteinGym using Spearman correlation (higher is better). Taxonomic clustering is evaluated at five hierarchical levels—class, order, family, genus, and species—using homogeneity scores, with higher values indicating more taxonomically consistent clusters. Additional clustering metrics are reported in Appendix A.4.

| Model | TreeBase ↓ | TreeFam ↓ | ProteinGym ↑ | Class ↑ | Order ↑ | Family ↑ | Genus ↑ | Species ↑ |
|---|---|---|---|---|---|---|---|---|
| PHYLA-MLM | 0.83 | 0.79 | 0.55 | 0.66 | 0.76 | 0.80 | 0.87 | 0.95 |
| PHYLA-NoAttention | 0.86 | 0.78 | 0.44 | 0.69 | 0.74 | 0.78 | 0.82 | 0.92 |
| PHYLA | **0.73** | **0.58** | **0.64** | **0.71** | **0.76** | **0.87** | **0.93** | **0.98** |

framework in biology, delineating evolutionary relationships between organisms and serving as an indicator of relative phenotypic traits. Current approaches to constructing the tree of life typically rely on multiple sequence alignments of ribosomal proteins (Hug et al., 2016). We used PHYLA to analyze a set of 3,084 phylogenetic sequences, successfully reconstructing the tree of life in just 16 hours, compared to the 3,840 hours required by traditional methods (Hug et al., 2016) (Appendix A.2).

In order to evaluate the validity of the reconstructed tree of life, we compare the tree to the multiple sequence alignment and perform manual feature inspection. As shown in Figure 1, PHYLA accurately places sequences within their respective domains in the tree of life. PHYLA identifies overlap between certain Archaeal isolates and Bacteria, a result consistent with current phylogenetic reasoning. Lokiarchaeota, an Archaeal lineage clustered with Bacteria, is known to have a mosaic genome with over 30% of its genome derived from Bacteria (Levasseur et al., 2017). Within this genus, PHYLA placed Lokiarchaeaota archaeon loki (L-A) paraphyletic to Bacteria while Lokiarchaeota 45 8 (L-45) is paraphyletic to Archaea (Figure 4a). Examination of the multiple sequence alignment of L-45 and L-A with their immediate phylogenetic neighbors revealed that L-45 harbors a deletion of the S3 ribosomal protein while L-A retains this protein (Figure 4b). The S3 deletion has been noted in

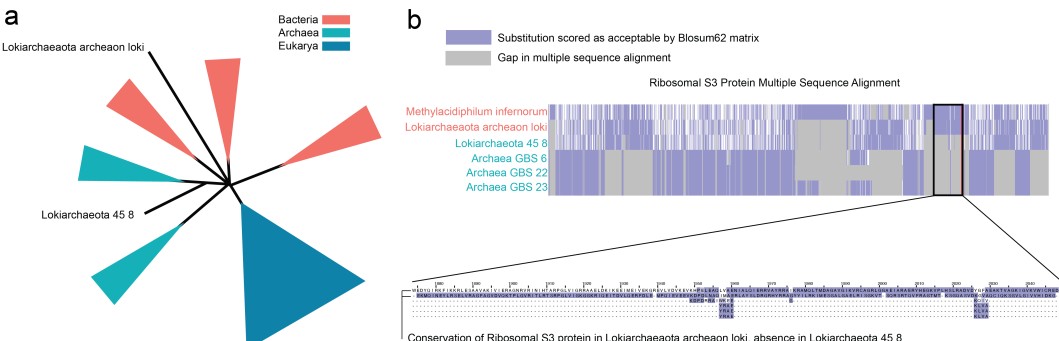

Figure 4: **PHYLA created a new placement of Lokiarchaea.** (a) Lokiarchaeaota archeaon loki (L-A) was placed among Bacterial neighbors while Lokiarchaeota 45 8 (L-45) was placed among Archaeal neighbors. (b) Analysis of the multiple sequence alignment revealed that L-A placed with Bacteria retained a conserved S3 ribosomal protein, aligning with its Bacterial neighbors. In contrast, the L-45 placed with Archaea exhibited a deletion of the S3 ribosomal protein, aligning with its Archaeal neighbors.

previous studies of Lokiarchaea genomes Da Cunha et al. (2017). Biologically, these differences may relate to adaptation to extreme environments. L-45 was isolated from the bottom of the Arctic Ocean, while L-A was isolated from the Horonobe Underground Research Laboratory (URL) in Japan. In fact, L-A's neighbor, Methylacidiphilum infernorum, is an acidophilic methanotroph originally isolated from a geothermal area in New Zealand Hou et al. (2008). This environment shares similarities with the conditions in the URL, where extensive methane metabolism has been observed Amano et al. (2024). This highlights PHYLA's ability to discover potentially biologically meaningful evolutionary relationships through probing the output reconstructed tree. We also apply PHYLA to whole-genome sequences of Mycobacterium tuberculosis, aligning over 4 megabases of DNA to infer novel phylogenetic relationships among clinical isolates (Appendix A.3).

## 6  Conclusion

We introduced an evolutionary reasoning benchmark showing that existing PLMs fail to recover phylogenetic structure, revealing a gap between sequence modeling and evolutionary reasoning. To address this, we proposed PHYLA, a hybrid state-space transformer trained with a tree-based objective. PHYLA achieves state-of-the-art performance on evolutionary reasoning and competitive accuracy on functional prediction, despite using far fewer parameters and data than large PLMs.

Traditional alignment-based methods such as MAFFT + FastTree remain the accuracy benchmark, but PHYLA narrows this gap while offering orders-of-magnitude faster inference. Limitations include reliance on distance-based tree reconstruction, operation in Euclidean space that may underrepresent hierarchical structure, and an evolutionary benchmark that, while practical, cannot fully capture the diversity or open-ended nature of protein evolution. As more curated phylogenetic datasets become available, expanding this benchmark is an important next step. Overall, PHYLA shows that evolutionary reasoning does not emerge from standard sequence-modeling paradigms and points toward a new class of models that learn evolution directly.

## Acknowledgments and Disclosure of Funding

Y.E. is supported by grant T32 HG002295 from the National Human Genome Research Institute, the NSDEG fellowship, and the Eric and Wendy Schmidt Fellowship. A.S. is supported by NSF GRFP grant DGE-2146755. Y.E. and M.Z. gratefully acknowledge the support of NIH R01-HD108794, NSF CAREER 2339524, US DoD FA8702-15-D-0001, awards from Harvard Data Science Initiative, Amazon Faculty Research, Google Research Scholar Program, AstraZeneca Research, Roche Alliance with Distinguished Scientists, Sanofi iDEA-iTECH Award, Pfizer Research, Chan Zuckerberg Initiative, John and Virginia Kaneb Fellowship award at Harvard Medical School, Biswas Computational Biology Initiative in partnership with the Milken Institute, and Kempner Institute for the Study of Natural and Artificial Intelligence at Harvard University. Any opinions, findings, conclusions or

recommendations expressed in this material are those of the authors and do not necessarily reflect the views of the funders.

**Competing Interests**

The authors declare no competing interests.

**Code Availability Statement**

Code to run PHYLA can be found in the project Github.

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

# A   Technical Appendices and Supplementary Material

Table 3: Information about the 1,533 TreeBase evaluation datasets.

| Dataset Metric | Min | Mean | Max | Standard Deviation |
|---|---|---|---|---|
| Average Sequence Length | 5 | 322 | 3658 | 288 |
| Number of Sequences | 4 | 53 | 1284 | 65 |

Table 4: Information about the 9,586 TreeFam evaluation datasets

| Dataset Metric | Min | Mean | Max | Standard Deviation |
|---|---|---|---|---|
| Average Sequence Length | 40 | 599 | 31926 | 687 |
| Number of Sequences | 4 | 33 | 375 | 34 |

## A.1   Tree reasoning and Protein Gym benchmark

### A.1.1   Tree reasoning benchmark

**Tree Reconstruction.** We evaluate tree reconstruction performance using two external datasets: TreeBase Piel & Tannen (2009) and TreeFam Li et al. (2006). TreeBase is a curated repository of 12,817 phylogenetic trees from peer-reviewed studies, along with associated sequence data contributed by users. After filtering for trees with complete and parseable sequence information, we retained 1,533 trees for analysis. TreeFam is a database of gene family trees focused on animal genomes. We extracted 9,586 gene trees along with their corresponding protein sequences. To evaluate reconstruction quality, we computed the normalized Robinson-Foulds distance (lower is better) between the predicted and reference trees. We found that PHYLA had the lowest normalized Robinson-Foulds distance among all models. This difference was significant ($p < 0.05$, one-sided t-test, Table 5) for all models.

We also evaluate the overlap between the training corpus (OpenProteinSet) and the evaluation sets (TreeBase, TreeFam). We ran BLAST Altschul et al. (1990) between each tree in TreeBase and TreeFam and all the sequences in the training corpus using an e-value cutoff of 1e-5 and bitscore cutoff of 50. We found in TreeBase only 0.56% of evaluation trees had any match to the training set. The overall mean percent identity of sequences in TreeBase is 0.38%. In TreeFam only 3.38% of evaluation trees had any match to the training set. The overall mean percent identity of sequences in TreeFam is 0.23%. These results indicate minimal leakage and highlights the ability of PHYLA to generalize.

Table 5: P-values from paired t-tests comparing PHYLA with baseline PLMs on normalized Robinson-Foulds scores for TreeBase and TreeFam. Lower values indicate statistically significant improvements.

| Baseline Model | TreeBase P-value | TreeFam P-value |
|---|---|---|
| ESM2 (650M) | $6.78 \times 10^{-16}$ | $1.54 \times 10^{-102}$ |
| ESMC (300M) | $1.69 \times 10^{-9}$ | $7.95 \times 10^{-222}$ |
| ESMC (600M) | $4.61 \times 10^{-27}$ | $7.16 \times 10^{-288}$ |
| ESM2 (3B) | $7.72 \times 10^{-24}$ | $3.95 \times 10^{-118}$ |
| ProGen2-Large (2.7B) | $1.37 \times 10^{-12}$ | $4.63 \times 10^{-112}$ |
| ProGen2-XLarge (6.4B) | $7.99 \times 10^{-99}$ | $1.37 \times 10^{-308}$ |
| ESM3 (1.4B) | $3.20 \times 10^{-40}$ | $5.41 \times 10^{-249}$ |
| Evo2 (7B) | $4.71 \times 10^{-65}$ | $1.37 \times 10^{-308}$ |

**Taxonomic Clustering Evaluation.** We evaluate taxonomic structure using data from the Genome Taxonomy Database (GTDB), a high-quality and frequently updated resource for microbial taxonomy. GTDB taxonomic labels are derived by concatenating multiple sequence alignments of 120 marker

proteins per organism. We use Release 10-RS226 (released April 16, 2025), which contains 715,230 organisms.

To construct our evaluation dataset, we randomly sample 50 groups of 10 organisms each, ensuring that all organisms within a group share the same taxonomic label at either the class, order, family, genus, or species level depending on the benchmark. For each model, we compute organism-level embeddings as follows: (1) For baseline models, we embed each of the 120 marker proteins independently, then average the resulting embeddings to produce a single vector per organism. (2) For PHYLA, we batch all 500 organisms for each marker protein (e.g. protein 1) and compute embeddings jointly. We repeat this for all 120 proteins, then average across them to obtain organism-level embeddings. We then perform k-means clustering (as implemented in scikit-learn) with k=50, corresponding to the number of groups.

In the main text, we report results for one such sample of 500 organisms. However, we find that performance is consistent across clustering metrics (completeness, homogeneity, NMI) and across random seeds. We generate four additional random samples and report mean and standard deviation across these replicates. PHYLA achieves state-of-the-art performance in taxonomic clustering across all taxonomic levels. It is only outperformed by ProGen2-XLarge in completeness at the class level (Table 6).

### A.1.2   Extended Benchmarking

**Fine-tuned ESM2-650M.** We fine-tuned the strongest baseline model, ESM2-650M, using two lightweight heads—a feed-forward network and a transformer head—trained with the same quartet loss as PHYLA, while freezing the backbone (as backpropagation through 650M parameters for trees with more than 100 leaves is computationally infeasible). Despite this targeted fine-tuning, the best variant achieved normalized Robinson–Foulds (normRF) scores of 0.77 on TreeBase and 0.79 on TreeFam, whereas PHYLA —trained end-to-end to reason jointly over sets of sequences—achieved 0.73 and 0.58, respectively. These results indicate that the performance gap cannot be attributed solely to evaluating PLMs in a zero-shot setting.

**Input-Order Robustness.** For each evaluation tree in TreeBase and TreeFam, PHYLA processes the full set of sequences in a single forward pass. Because the model is trained on randomly sampled and permuted sequence subsets, its predictions are invariant to input order. To verify this property, we randomly shuffled the sequence order for 250 trees from TreeBase five times each and measured the standard deviation in normalized RF scores. The mean deviation was 0.01, confirming that PHYLA's embeddings and resulting distance matrices are effectively deterministic with respect to sequence order.

**Training signal comparison.** To assess whether the choice of supervision signal influences performance, we compared two variants of PHYLA trained using (i) normalized Hamming distances computed from multiple sequence alignments (MSAs) and (ii) pairwise patristic distances extracted from trees reconstructed with FastTree on the same MSAs. Both variants achieved nearly identical performance on TreeBase and TreeFam benchmarks, with differences below 0.01 in normalized Robinson–Foulds distance (Table 7).

These results indicate that PHYLA does not simply replicate a specific distance metric but learns generalizable representations that capture consistent topological structure across distance definitions. In practice, Hamming distances serve as an efficient proxy for topological supervision without materially affecting model behavior or accuracy.

### A.1.3   ProteinGym benchmark

The 83 ProteinGym datasets were chosen based on the memory constraint of a single 80GB H100 GPU. We stratified performance on the ProteinGym functional prediction benchmark across various levels of overlap between pre-training and evaluation datasets. We quantified overlap by running the Basic Local Alignment Search Tool (BLAST) algorithm between each representative sequence of the 83 ProteinGym datasets and each model's pre-training dataset (Altschul et al. (1990)). Overlap was calculated as the average percent similarity of output hits. We did not include the total number of hits when calculating the overlap metric in order to not further penalize models with larger pre-training datasets.

Table 6: Clustering evaluation metrics (mean ± std) across taxonomic levels. Standard deviation and mean across 5 random taxonomic group samples from GTDB.

| Taxonomic Level | Model | Homogeneity | Completeness | NMI |
|---|---|---|---|---|
| Class | ESM2 (650M) | 0.54 ± 0.05 | 0.64 ± 0.04 | 0.58 ± 0.05 |
| | ESM3 (1.4B) | 0.67 ± 0.01 | 0.71 ± 0.01 | 0.69 ± 0.01 |
| | ESMC (300M) | 0.55 ± 0.10 | 0.62 ± 0.10 | 0.58 ± 0.10 |
| | ESMC (600M) | 0.51 ± 0.02 | 0.59 ± 0.02 | 0.55 ± 0.02 |
| | ProGen2-Large (2.7B) | 0.67 ± 0.02 | 0.74 ± 0.02 | 0.70 ± 0.02 |
| | ProGen2-XLarge (6.4B) | 0.66 ± 0.04 | **0.75 ± 0.03** | 0.70 ± 0.03 |
| | Evo2 (7B) | 0.58 ± 0.02 | 0.67 ± 0.01 | 0.62 ± 0.01 |
| | PHYLA (24M) | **0.69 ± 0.02** | 0.73 ± 0.02 | **0.71 ± 0.02** |
| Family | ESM2 (650M) | 0.57 ± 0.05 | 0.68 ± 0.04 | 0.62 ± 0.04 |
| | ESM3 (1.4B) | 0.75 ± 0.02 | 0.78 ± 0.03 | 0.76 ± 0.02 |
| | ESMC (300M) | 0.59 ± 0.11 | 0.67 ± 0.10 | 0.63 ± 0.11 |
| | ESMC (600M) | 0.48 ± 0.04 | 0.61 ± 0.04 | 0.54 ± 0.04 |
| | ProGen2-Large (2.7B) | 0.74 ± 0.04 | 0.81 ± 0.03 | 0.77 ± 0.03 |
| | ProGen2-XLarge (6.4B) | 0.65 ± 0.07 | 0.76 ± 0.05 | 0.70 ± 0.06 |
| | Evo2 (7B) | 0.62 ± 0.05 | 0.72 ± 0.05 | 0.67 ± 0.05 |
| | PHYLA (24M) | **0.85 ± 0.02** | **0.88 ± 0.02** | **0.86 ± 0.02** |
| Genus | ESM2 (650M) | 0.64 ± 0.05 | 0.74 ± 0.04 | 0.68 ± 0.05 |
| | ESM3 (1.4B) | 0.83 ± 0.02 | 0.85 ± 0.02 | 0.84 ± 0.02 |
| | ESMC (300M) | 0.65 ± 0.10 | 0.73 ± 0.11 | 0.69 ± 0.10 |
| | ESMC (600M) | 0.51 ± 0.09 | 0.66 ± 0.06 | 0.58 ± 0.08 |
| | ProGen2-Large (2.7B) | 0.77 ± 0.06 | 0.85 ± 0.04 | 0.81 ± 0.05 |
| | ProGen2-XLarge (6.4B) | 0.68 ± 0.08 | 0.79 ± 0.05 | 0.73 ± 0.07 |
| | Evo2 (7B) | 0.74 ± 0.06 | 0.80 ± 0.04 | 0.77 ± 0.05 |
| | PHYLA (24M) | **0.95 ± 0.02** | **0.97 ± 0.01** | **0.96 ± 0.01** |
| Order | ESM2 (650M) | 0.57 ± 0.03 | 0.67 ± 0.03 | 0.62 ± 0.03 |
| | ESM3 (1.4B) | 0.72 ± 0.03 | 0.76 ± 0.02 | 0.74 ± 0.03 |
| | ESMC (300M) | 0.58 ± 0.10 | 0.65 ± 0.10 | 0.61 ± 0.10 |
| | ESMC (600M) | 0.51 ± 0.03 | 0.63 ± 0.02 | 0.57 ± 0.03 |
| | ProGen2-Large (2.7B) | 0.68 ± 0.04 | 0.76 ± 0.04 | 0.72 ± 0.04 |
| | ProGen2-XLarge (6.4B) | 0.65 ± 0.03 | 0.75 ± 0.03 | 0.69 ± 0.03 |
| | Evo2 (7B) | 0.60 ± 0.05 | 0.69 ± 0.03 | 0.64 ± 0.04 |
| | PHYLA (24M) | **0.78 ± 0.03** | **0.81 ± 0.02** | **0.79 ± 0.02** |
| Species | ESM2 (650M) | 0.69 ± 0.05 | 0.79 ± 0.04 | 0.73 ± 0.05 |
| | ESM3 (1.4B) | 0.85 ± 0.03 | 0.88 ± 0.03 | 0.87 ± 0.03 |
| | ESMC (300M) | 0.73 ± 0.06 | 0.82 ± 0.02 | 0.77 ± 0.04 |
| | ESMC (600M) | 0.52 ± 0.04 | 0.72 ± 0.04 | 0.60 ± 0.04 |
| | ProGen2-Large (2.7B) | 0.85 ± 0.04 | 0.90 ± 0.03 | 0.87 ± 0.03 |
| | ProGen2-XLarge (6.4B) | 0.73 ± 0.04 | 0.83 ± 0.03 | 0.77 ± 0.03 |
| | Evo2 (7B) | 0.82 ± 0.03 | 0.87 ± 0.03 | 0.84 ± 0.03 |
| | PHYLA (24M) | **0.98 ± 0.01** | **0.99 ± 0.00** | **0.99 ± 0.01** |

When comparing functional prediction performance between PHYLA and ESM2 on all 83 ProteinGym datasets, ESM2 significantly outperforms PHYLA by 0.14. However, on low-overlap regimes with less than 40% similarity to the model's pre-training dataset, PHYLA outperforms ESM2 by 0.03 as shown in Table 8 (Pearson (2013). Performance degrades on low-overlap regimes for all models, but PHYLA's performance degrades less. In addition, PHYLA occupies a lower overlap regime than ESM2, which is likely due to the smaller pre-training set used for PHYLA.

Table 7: Performance comparison when training with Hamming versus tree-based supervision signals.

| Training Signal | TreeBase (normRF) | TreeFam (normRF) |
|---|---|---|
| Hamming Distance (original) | 0.73 | 0.58 |
| FastTree Distance | 0.73 | 0.58 |

Table 8: **Functional Prediction Across Various Overlap Settings.** Model performance on predicting functional effects in ProteinGym stratified into low-overlap and high-overlap settings based on alignment overlap.

| Model | Low-Overlap | High-Overlap | All Datasets |
|---|---|---|---|
| ESM2 | 0.59 | **0.80** | **0.78** |
| PHYLA-MLM | 0.53 | 0.61 | 0.55 |
| PHYLA-NoAttention | 0.40 | 0.53 | 0.44 |
| PHYLA | **0.62** | 0.68 | 0.64 |

## A.2 Tree of Life analysis

To reconstruct the tree of life, we use a dataset of 3,083 organisms, each represented by the concatenated multiple sequence alignment (MSA) of 16 conserved ribosomal proteins Hug (2016). For input to PHYLA, we concatenate the raw amino acid sequences of all 16 ribosomal proteins per organism, yielding sequence lengths of approximately 6 million tokens. We process these sequences using PHYLA on a high-memory CPU machine (800 GB RAM), requiring approximately 16 hours of compute time for a full forward pass. We then compute pairwise distances between organism-level embeddings and apply the neighbor-joining algorithm to construct the resulting phylogenetic tree.

## A.3 Tuberculosis analysis

Tuberculosis (TB) is one of the leading causes of death globally due to infectious disease World Health Organization (2024). A major challenge in controlling TB is its capacity to rapidly evolve drug resistance. Genomic surveillance has become a critical tool in addressing this issue: by tracking the evolutionary dynamics of TB, we can detect emerging resistance before it becomes widespread and adapt treatment strategies accordingly Thorpe et al. (2024). We use PHYLA to construct the first whole-genome phylogenetic tree of *Mycobacterium tuberculosis* isolates. Specifically, we analyze 151 complete TB genomes (around 4 Mb each) Marin et al. (2025). To generate organism-level representations, we divide each genome into non-overlapping 500 bp segments (shared across all 151 genomes), embed each chunk using PHYLA, and then average the resulting embeddings across the genome. This approach is well-suited to TB, which is a highly clonal organism with limited variation across isolates—making direct genome chunking and alignment across samples both feasible and meaningful Freschi et al. (2021). We compute pairwise distances between the organism-level embeddings and reconstruct the phylogenetic tree using the neighbor-joining algorithm (Figure 5).

PHYLA accurately reconstructs the major global lineages of *Mycobacterium tuberculosis*, recovering coherent clades corresponding to Lineages 1–6 and the recently described Lineage 8 Freschi et al. (2021). The topology is broadly consistent with established TB phylogenies, with early-branching Lineage 1 and distinct clusters for the derived East Asian (Lineage 2) and Euro-American (Lineage 4) groups. Deviations are observed in the relative positioning of Lineages 3 and 4, which appear slightly closer than expected. Given that PHYLA infers these relationships directly from sequence embeddings rather than explicit substitution models, the recovery of lineage-level structure without supervision underscores that the learned representation captures genuine evolutionary signal beyond surface sequence similarity.

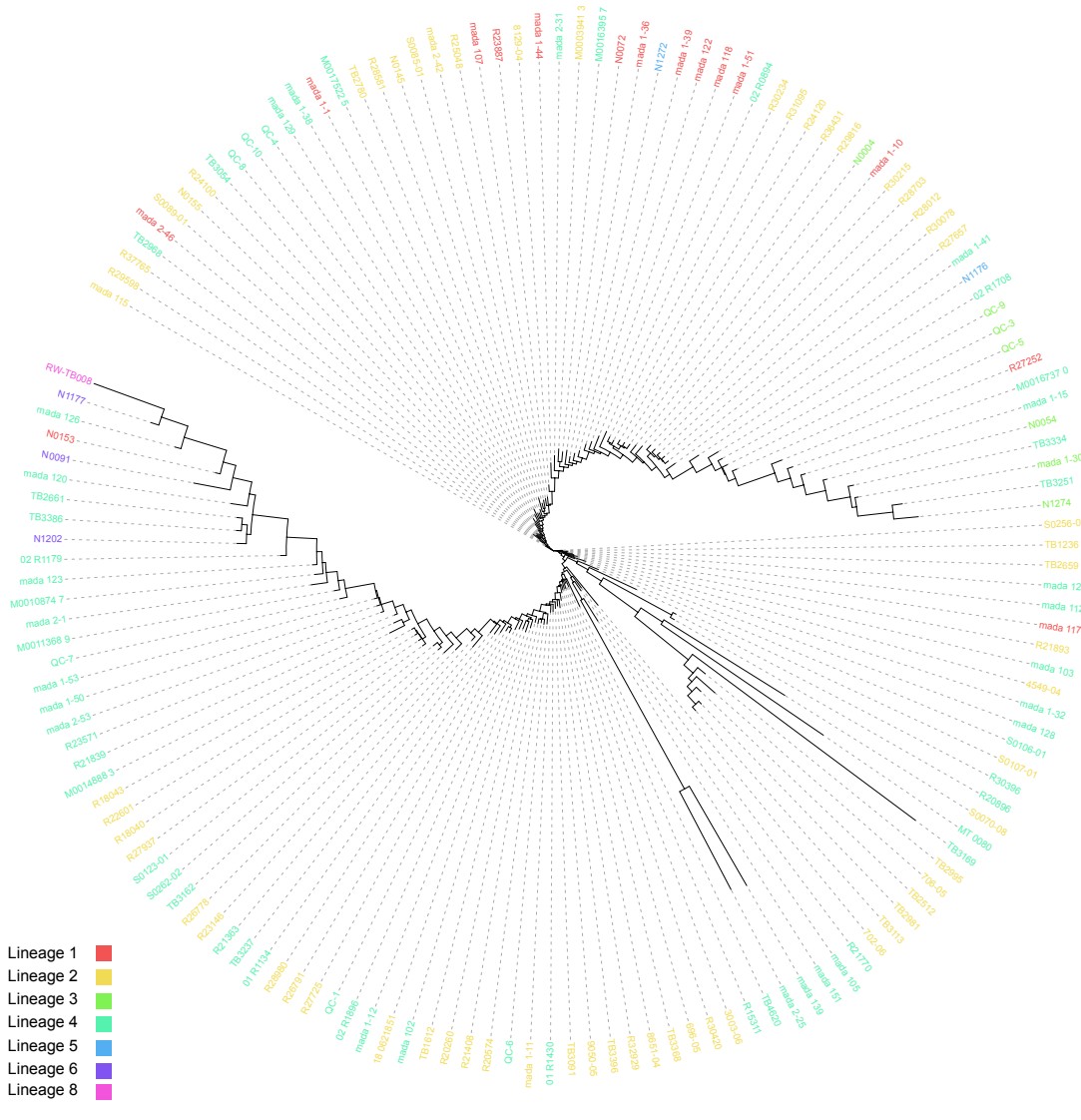

Figure 5: **PHYLA-generated tree of whole genome Tuberculosis (TB) isolates.** By comparing chunks of the genome of TB across 151 TB isolates, PHYLA constructed the first whole genome tree of TB. Labeled are the main lineages of TB (lineage2, lineage 3, and lineage 4).

## A.4   Ablations

We retested our ablations, PHYLA-MLM trained with only the MLM loss and PHYLA-NoAttention trained without sparsified attention, on the extended taxonomic clustering dataset described in Section A.1.1. We found across all taxonomic levels, except class, PHYLA had the best performance across all metrics assessed. PHYLA also has significantly better tree reconstruction on TreeBase and TreeFam than all ablations (Table 10).

Table 9: Clustering evaluation metrics (mean ± std) for PHYLA variants across taxonomic levels. Standard deviation and mean across 5 random taxonomic group samples from GTDB.

| Taxonomic Level | Model | Homogeneity | Completeness | NMI |
|---|---|---|---|---|
| Class | PHYLA-MLM | 0.65 ± 0.02 | 0.69 ± 0.02 | 0.67 ± 0.02 |
| | PHYLA-NoAttention | **0.69 ± 0.02** | **0.74 ± 0.01** | **0.71 ± 0.02** |
| | PHYLA | 0.69 ± 0.02 | 0.73 ± 0.02 | 0.71 ± 0.02 |
| Family | PHYLA-MLM | 0.78 ± 0.02 | 0.80 ± 0.02 | 0.79 ± 0.02 |
| | PHYLA-NoAttention | 0.79 ± 0.02 | 0.83 ± 0.02 | 0.81 ± 0.02 |
| | PHYLA | **0.85 ± 0.02** | **0.88 ± 0.02** | **0.87 ± 0.02** |
| Genus | PHYLA-MLM | 0.89 ± 0.02 | 0.92 ± 0.02 | 0.90 ± 0.02 |
| | PHYLA-NoAttention | 0.85 ± 0.03 | 0.88 ± 0.02 | 0.87 ± 0.03 |
| | PHYLA | **0.95 ± 0.02** | **0.97 ± 0.01** | **0.96 ± 0.01** |
| Order | PHYLA-MLM | 0.74 ± 0.01 | 0.78 ± 0.01 | 0.76 ± 0.01 |
| | PHYLA-NoAttention | 0.74 ± 0.03 | 0.79 ± 0.02 | 0.76 ± 0.02 |
| | PHYLA | **0.78 ± 0.03** | **0.81 ± 0.02** | **0.79 ± 0.02** |
| Species | PHYLA-MLM | 0.94 ± 0.01 | 0.96 ± 0.01 | 0.95 ± 0.01 |
| | PHYLA-NoAttention | 0.92 ± 0.01 | 0.94 ± 0.01 | 0.93 ± 0.01 |
| | PHYLA | **0.98 ± 0.01** | **0.99 ± 0.00** | **0.99 ± 0.01** |

Table 10: P-values from paired t-tests comparing PHYLA with PHYLA ablations on normalized Robinson-Foulds scores for TreeBase and TreeFam. Lower p-values indicate stronger statistical significance.

| Baseline Model | TreeBase P-value | TreeFam P-value |
|---|---|---|
| PHYLA-MLM | $1.14 \times 10^{-57}$ | 0.0 |
| PHYLA-NoAttention | $1.94 \times 10^{-91}$ | 0.0 |

## A.5 Evolution Reasoning Result

Please see Section A.1.1.

## A.6 Recent Work

**Evolutionary reasoning versus evolutionary modeling.** Previous studies distinguish between fitting a generative model to observed sequence data and reasoning about the underlying fitness landscape that produced those sequences Weinstein et al. (2022); Ding et al. (2019). From these works, we know generative sequence models aim to match the marginal distribution of extant proteins (evolutionary modeling), whereas evolutionary reasoning seeks to recover the selective pressures and functional constraints driving those distributions.

**AI for Phylogenetic Tree Construction.** Recent years have seen a surge of AI-driven approaches for evolutionary-biology problems. For instance, PhyloVAE and ARTree Xie et al. (2025); Xie & Zhang (2023) generate plausible tree topologies from collections of existing phylogenies. GeoPhy and Phyloformer Mimori & Hamada (2023); Nesterenko et al. (2025) take a multiple sequence alignment (MSA) as input and infer both tree topology and branch lengths by maximizing the likelihood under a fixed substitution model. Other methods—such as DEPP and DeePhy Jiang et al. (2022); Mahapatra & Mukherjee (2025)—are designed to place unaligned query sequences into a reference phylogenetic tree or into existing triplets of sequences. In contrast, PHYLA focuses on a task that combines these capabilities: starting from unaligned sequences alone, it simultaneously infers a complete tree topology without requiring any prior alignment.

### A.7 Evaluation on Masked Token Prediction

In addition to evaluating the ability of PHYLA and benchmark models to perform evolutionary reasoning, we also assess performance on a standard PLM task of masked-token prediction. PHYLA trained only with masked-language modeling attains 30% top-1 accuracy with reduced evolutionary reasoning performance; training with tree loss decreases accuracy to 11% but improves evolutionary reasoning performance (See PHYLA-MLM performance in 2 for evolutionary reasoning performance). Conventional PLMs (ESM-2 650M, ProGen2) achieve 45–55%. These results confirm a trade-off: per-token objectives favor single-sequence reconstruction, whereas the tree loss preserves cross-sequence signals essential for evolutionary reasoning.

### A.8 Societal Impacts and Safety Concerns

Our method's ability to generalize beyond its training distribution can lead to more accurate identification of disease-causing variants in previously unseen protein families, potentially accelerating both basic research and the development of personalized therapeutics. However, because this is the first model that infers a phylogenetic tree directly from raw sequence data, there is a risk that users may overestimate its reliability and substitute it for established, alignment-based pipelines. Such overreliance could produce misleading evolutionary hypotheses or clinical interpretations if the model's assumptions and limitations are not carefully considered. To mitigate these dangers, we stress that our approach is intended as a complement to—rather than a replacement for—traditional phylogenetic methods, and that any high-stakes decisions should always include orthogonal validation.

