# OpenReview forum: "Evolutionary Reasoning Does Not Arise in Standard Usage of Protein Language Models"
_NeurIPS.cc/2025/Conference — NeurIPS 2025 poster_

### Official Review · Reviewer_Ya7G · 2025-06-30

**Clarity:** 3
**Significance:** 4
**Originality:** 4
**Rating:** 5
**Confidence:** 4

**Summary:**

This model proposed an evaluation benchmark to exam the existing PLMs on phylogenetic reconstruction and a new model PHYLA for evolutionary reasoning. The new model rely on a hybrid state-space and attention-based architecture to capture both long-range dependency and local context. It can process multiple sequences in parallel to discover inter and intra sequence relationship. The model is trained on a tree loss function to reflect evolutionary distance and phylogenetic structure in the embedding.
Compared to existing PLMs, who often failed on phylogenetic reconstruction, PHYLA achieved good accuracy and efficiency on phylogenetic inference on benchmark datasets.
This method is effective on evolutionary inference without sequence alignment and guide tree.

**Questions:**

1. It would be interesting to discuss about why PLMs can’t even beat hamming distance on phylogenetic inference. That means PLMs does not capture basic evolutionary information from the sequences.
2. How does the model performance compare to traditional phylogenetic inference methods? On both accuracy, running time and memory usage.

**Ethical Concerns:**

["NO or VERY MINOR ethics concerns only"]

**Final Justification:**

Thanks for the authors' detailed rebuttal and the new experiments. They have addressed my concerns.
Besides, I disagree with the comments of reviewer 9GvF. The concerns this reviewer raised was based on a fundamental misunderstanding of the core contribution and misinterpretation of the experiment results. In my point of view, this work is a solid and impactful work that deserved to be accepted.
Regarding to the problem with Euclidean distance, the authors' entire point is that even with a standard and widely used metric like Euclidean distance, PLMs still fail to produce meaningful evolutionary representations. The failure isn't in the evaluation method but in the model itself.
Reviewer 9GvF also raised another concern that this new method is not good enough because it does not beat the FastTree. This totally missed the point. The author has already included FastTree as a critical baseline to provide a performance upper bound. I would suggest that the author make it clear that their methodology are completely different. Traditional methods like FastTree rely heavily on MSAs, which is extremely computational expensive as tree size grow. The key contribution of this work is developed an efficient and scalable alternative and it is alignment-free. This isn't about simply beating an existing method, it is about introduce a pioneer approach for evolutionary reasoning.
I felt that the reviewer 9GvF is not familiar with phylogenetic inference background and I totally understand their concern on the performance side.
I believe the authors have already made a strong and convincing argument. This work is a significant contribution to the field. I recommend for acceptance.

**Limitations:**

Overall, this paper is a solid study on a biologically significant problem, phylogenetic inference. This study not only identified an astonishing fact that the existing PLMs are not applicable on evolutionary studies, but also proposed a solid solution to resolve this problem by introducing both intra and inter sequence inference blocks, tree loss to capture tree information directly from the model.
The limitation of this work mainly include leak of traditional phylogenetic inference baselines, model robustness analysis, and theoretical analysis on the model effectiveness.

**Quality:**

3

**Strengths And Weaknesses:**

This paper reveals an important question that the evolutionary reasoning is different from sequence modeling. Evaluation on existing PLMs shows that the limitation of large language models on specific biological tasks. The impact is huge for comparative studies.
The design of model combines state-space module with sparsified attention mechanism. Effectively capture both intra and inter sequence relationships. The tree loss enables the model to learn evolutionary relationships directly from sequences, which is quite novel and biologically significant.
The model can process multiple sequence semutaniousy, infer phylogenetic tree without multiple sequence alignment and guide tree, greatly improved inference efficiency.
The experiment is comprehensive, covers multiple tasks, include multiple benchmark datasets.
Some weaknesses include that this study does not include traditional phylogenetic tree inference, such as RAxML, Fasttree, etc. It is unclear about the model robustness and effectiveness on datasets with high sequence diversity. It would be helpful to include simulated datasets with artificially designed sequence diversity to simulate sequences with different evolutionary distance.

---

> ### Author Rebuttal · Authors · 2025-07-31
>
> # Question 1
>
> We thank the reviewer for highlighting this important point. Conventional PLMs are trained with a masked-language-model (MLM) loss on single sequences. That objective optimises per-token reconstruction, not the relationships among sequences, so the embeddings do not preserve evolutionary distance. A simple Hamming count is crude but is monotonically related to divergence, so outperforms these models on tree reconstruction.
>
> Our experiments make this explicit. Swapping PHYLA’s quartet-based tree loss for MLM decreases the ability for PHYLA to do evolutionary reasoning; fine-tuning a head on the 650M-parameter ESM-2 backbone likewise fails to beat Hamming, and removing PHYLA’s multi-sequence attention blocks drops performance dramatically. Scale cannot rescue a model whose objective and architecture is misaligned with the task.
>
> This result matters for the community. It shows that objective and architecture alignment, not parameter count, is the bottleneck, and it motivates our Evolutionary-Reasoning Benchmark and PHYLA’s design. Publishing these findings will prevent researchers from assuming that large MLM-trained PLMs “implicitly learn evolution” and will guide future work toward loss functions and architectures that capture the phylogenetic signal. PHYLA represents the first step in a new class of evolutionary-focused PLMs with its own unique model architectures, loss functions, and benchmarks.
>
> # Question 2
>
> We thank the reviewer for raising this point. Our original focus was to test whether general‑purpose PLMs capture phylogenetic structure, but we agree that including a classical pipeline provides an essential upper bound. We have therefore run the standard MAFFT + FastTree workflow on the same TreeBase and TreeFam splits. It achieves the best tree‑reconstruction scores (normRF ≈ 0.65 on TreeBase and ≈ 0.32 on TreeFam), while PHYLA attains 0.73 and 0.58 respectively—closing roughly half of the gap that separates PLMs from classical methods. Importantly, the classical run required ~2 hours on TreeBase and ~66 hours on TreeFam across multiple CPU nodes (30 nodes with 75 GB memory each with  52% usage each), whereas PHYLA produced both trees in under an hour on a single H100 GPU (83% memory usage). These results, now reported in the appendix, clarify the landscape: classical align‑then‑tree pipelines remain the accuracy benchmark, but PHYLA offers a substantial speed advantage and demonstrates that models can approach SOTA accuracy without an explicit alignment step.

---

### Official Review · Reviewer_9GvF · 2025-07-02

**Clarity:** 2
**Significance:** 2
**Originality:** 2
**Rating:** 2
**Confidence:** 4

**Summary:**

This work's main contribution is a novel tree-based loss function for (pre-)training PLMs, which is crucially distinct from MLM. The work trains a new multi-sequence model called PHYLA using this loss. Thanks to this loss, PHYLA's latent space better reflects true tree evolutionary distances. Using PHYLA's latent space improves performance on phylogenetic tree reconstruction, compared to using the embeddings of traditional PLMs.

**Questions:**

1) In the appendix, line 586, why do you state that tree reconstruction could introduce biases during training? You are using the Hamming distance, is that not "more biased"? To my understanding, this essentially contradicts the whole body of work on statistical phylogenetics.
2) Why do you use the Euclidean distance on the embedding space of traditional PLMs? Might fitting a network to "correct" potential "nonlinearities" in these embeddings make them perform well on the phylogenetic tree reconstruction task? This would put into question the claim that traditional PLMs are not well positioned to perform evolutionary reasoning.
3) Is it possible that there is test set leakage in the phylogenetic tree reconstruction task? I.e., how do OpenProteinSet and TreeBase/TreeFam overlap?
4) If PHYLA was trained on aligned sequences, why is there any expectation that it would generalize to unaligned sequences?

**Ethical Concerns:**

["NO or VERY MINOR ethics concerns only"]

**Final Justification:**

I find the rebuttal of the authors is still not convincing. Moreover, the fact that the authors confirmed that important information was originally hidden in the appendix (specifically, the fact that Euclidean distance in embedding space is used to evaluate PLMs) further decrease my confidence in the submission, so I continue to recommend rejection. I understand that reviewer Ya7G enthusiastically recommends the paper for acceptance but I disagree with them. The reviewer has mentioned in their review as a strength the "astonishing fact that the existing PLMs are not applicable on evolutionary studies". Both myself and reviewer 8BeB have found this unsurprising (in fact, trivial) because of how PLMs are not trained to explicitly match tree distances, unlike PHYLA. I disagree with reviewer Ya7G, and am surprised by their high confidence in their score. I think the quartet loss is interesting, but as marketed the paper is making strong claims about the limitations of PLMs which I am still not convinced by, and, to reiterate, the fact that important information was indeed deferred to the appendix reduces my confidence in the quality of the submission. I have maintained my score of 2 and increased my confidence from 3 to 4.

UPDATE: The authors have promised to overhaul their paper to remove their strong claims about the limitations of PLMs. Please see my "Second Response to Authors" and the author's final response regarding this. In particular, this involves changing the title of the paper. I agree that with the proposed changes the paper would be of interest, but since the changes with respect to the original submission are so substantial, it is hard for me to confidently recommend acceptance without reading the promised version, so I will maintain my score of 2 recommending rejection, but I am open to discussion.

**Limitations:**

As mentioned in my review, many important facts -- specifically, limitations -- are deferred to the appendix and not clearly communicated and discussed upfront in the main text.

**Paper Formatting Concerns:**

No concerns.

**Quality:**

2

**Strengths And Weaknesses:**

Stregths:
1) The tree-based loss function is interesting and clearly distinct from MLM. It reminds me of contrastive losses which try to  to place closely related objects closer in latent space. These kinds of losses have seen success in other areas such as computer vision.

Weaknesses:
1) Firstly, the way in which PLMs are used in the tree reconstruction benchmark is by just taking the Euclidean distance in embedding space. This fact deferred to the appendix (line 607) and is quite important: none of these PLMs (e.g. ESM) have been trained to recapitulate true tree distance, so in a sense it is expected that they will not do well on phylogenetic tree reconstruction -- at least not as well as PHYLA, since PHYLA has been trained to recover quartets. There could be non-linear effects on the latent space of traditional PLMs, but if you trained a network to "correct" these embeddings such that they recapitulate tree distance (using, for example, your quartet-based tree loss, or by other means such as matching to true tree distances), the corrected embeddings might do just as well as PHYLA on tree reconstruction. I thus find the presented results are unfairly harsh with current PLMs.
2) Secondly -- and also deferred to the appendix (line 585) -- is the fact that the model is trained using (normalized) Hamming distances instead of true tree distances, or more reallistically, distances obtained from reconstructed tree using software such as FastTree. The manuscript says that "To create trees we would have to run an algorithm on top of the MSA which could introduce biases while training". As someone with experience in statistical phylogenetics, I find this statement alarming and reversed: I would argue that using Hamming distances is biased whereas using tree reconstruction software such as FastTree or IQTree will give more accurate and less biased results. For example, Hamming distances disregard the charge of the residues, and in general are a non-linear transformation of true tree distance. Instead, CTMC-based software such as FastTree uses amino acid substitution rate matrices which account for residue charge and attempt to reconstruct true tree distance. Therefore, it appears to me that PHYLA is essentially learning to "imitate" the Hamming distance quartet resolution criteria, which seems an uninteresting task (compared to MLM, for example) via a continuous relaxation.
3) While PHYLA does not require aligned sequences, the fact that it was trained exclusively on MSAs (Section A.1.2: training details) makes me wonder why it would even generalize to unaligned sequences.
4) I didn't find any discussion on how the training set (OpenProteinSet) overlaps with the test set (TreeBase and TreeFam). If there is overlap, then there is test set leakage, and since PHYLA is tarined to explicitly recover quartets it would be unsurprising that it did well on the tree reconstruction task.

---

> ### Author Rebuttal · Authors · 2025-07-31
>
> # Weakness 1
>
> We thank the reviewer for this thoughtful and constructive comment. We agree that it is an important point that the way PLMs are used in the tree reconstruction benchmark is by simply computing Euclidean distances in embedding space (previously only mentioned in the appendix, line 607). We will move this detail to the main text to make the evaluation setting clearer.
> To address the concern that this setup may be unfairly harsh on current PLMs, we performed an additional experiment with the best-performing PLM on the benchmark, ESM-650M, by training a network on top of its embeddings using the same tree-loss employed by PHYLA. We froze ESM-650M during training, as it was not computationally feasible to store gradients for larger trees containing over 100 proteins; this limitation highlights a key advantage of PHYLA’s architecture, which is designed to jointly process all sequences efficiently.
>
> We evaluated two variants: (i) a simple feed-forward network (FNN) head and (ii) a transformer-based head over the protein embeddings. In both cases, ESM-650M was unable to match PHYLA’s performance. On the TreeBase benchmark, the corrected ESM-650M models achieved normalized RF scores starting at 0.78 and plateauing at 0.77/0.76, while on the TreeFam benchmark performance degraded to 0.79/0.80. These experiments show that PHYLA’s superior accuracy stems from its architecture’s ability to reason jointly over entire sequence sets, not from the PLMs being evaluated in zero‑shot settings. We have added these results to the appendix.
>
> # Weakness 2
>
> We thank the reviewer for highlighting this crucial point. Our intent was not to claim that normalized Hamming distance (HD) is less biased than tree‑based algorithms and distances, but rather to avoid injecting the hyper‑parameter choices and heuristic assumptions of a particular tree‑building algorithm into the supervision signal. We have clarified this wording in the paper and explicitly acknowledge that FastTree/IQ‑Tree (along with other algorithms, RaxML, etc) are state‑of‑the‑art in this space.
>
> ### Why HD was used originally.
>
> **Definition & clarity:** To be clear, the (normalized) HDs are computed column‑wise on each multiple‑sequence alignment (MSA) not on raw, unaligned sequences.
>
> **Practicality:** MSAs were already computed on the OpenProteinSet and HDs can be obtained from MSAs in O(L × N²) time, whereas building trees even with FastTree with a large number of trees is computationally expensive.
>
> **Focus on topology:** PHYLA’s quartet loss optimizes which branches merge, independent of branch length. FastTree likewise derives its initial topology from the pairwise similarity in the underlying MSA, then refines topology and branch lengths. We therefore treated HD as a proxy for topological signal, not a perfect metric of evolutionary time.
>
> ### New experiment requested by the reviewer.
>
> We reconstructed FastTree trees for all training MSAs, extracted pairwise tree distances, and re‑trained PHYLA with these distances in place of HD. Test results:
>
> | **Training Signal**     | **TreeBase** | **TreeFam** |
> |-------------------------|-------------:|------------:|
> | *HD (original)*         | **0.7269**   | **0.5777**  |
> | *FastTree Distance*     | **0.7312**   | **0.5777**  |
>
>
> The near‑identical scores indicate that PHYLA is not merely copying HD; it is learning to compare sequences in a way that generalizes across distance definitions.
>
> ### Take‑away.
>
> Using HD does not handicap PHYLA on the evaluated topological tasks, while avoiding dependence on specific FastTree/IQ‑Tree settings. We have added this explanation, the table above, and a note that incorporating richer substitution models for branch‑length prediction is valuable future work.
>
> # Weakness 3
>
> We thank the reviewer for raising this point. The current wording in Section A 1.2 is confusing, and we have revised it. To clarify, PHYLA is always fed raw, unaligned amino‑acid sequences. The only role of the multiple‑sequence alignments (MSAs) in training is to provide a target distance matrix—we compute normalized Hamming distances on the MSA, then discard the alignment. Thus the model never “sees” gap symbols and is explicitly trained on unaligned inputs.
>
> Because the supervision signal is derived from MSAs while the inputs are raw sequences, PHYLA must implicitly learn to align (or otherwise compare) sequences in a differentiable manner to minimise the quartet loss. This built‑in alignment‑free processing is precisely why the model generalises to novel, unaligned sequences at test time.
>
> # Weakness 4
>
> We thank the reviewer for pointing out the need to quantify any overlap between the training corpus (OpenProteinSet) and the evaluation sets (TreeBase, TreeFam). We ran BLAST between each tree in TreeBase and TreeFam and all the sequences in the training corpus using an e-value cutoff of 1e-5 and bitscore cutoff of 50. We found in TreeBase only 0.56% of evaluation trees had any match to the pretraining set. The overall mean percent identity of sequences in TreeBase is 0.38%. In TreeFam only 3.38% of evaluation trees had any match to the pretraining set. The overall mean percent identity of sequences in TreeFam is 0.23%. These results indicate minimal leakage and highlights Phyla’s ability to generalize. We will add this to the appendix and cite it in the main text.
>
> # Questions
>
> We thank the reviewer for raising these important questions, we believe our detailed responses above (the new experiments and clarifications) fully resolve these points.
>
> # Limitations
>
> We thank the reviewer for emphasizing that several limitations were in the appendix. Space constraints forced difficult trade‑offs, but we agree these details belong in the main narrative. In the revised draft we have moved all limitations from the appendix to the discussion and added more training details from the appendix to the main text. These changes make the scope and limitations explicit while respecting the page limit.

---

> > ### Author Response · Authors · 2025-08-06
> >
> > Dear Reviewer,
> >
> > Thank you again for your thoughtful initial review. We wanted to kindly follow up to see if you had any additional comments or thoughts after reading our rebuttal. The discussion phase ends soon, and we would greatly appreciate any updated feedback or clarifications you might have, especially regarding your score, concerns, or suggestions.
> >
> > We’ve tried to address all questions and weaknesses with additional experiments and clarifications. If anything remains unclear, we’d be happy to elaborate further.
> >
> > Many thanks again for your time and contributions to this process.
> >
> > Best regards,
> >
> > The Authors

---

> > ### Comment · Reviewer_9GvF · 2025-08-06
> > **Response to Authors**
> >
> > I thank the authors for their response. The authors have confirmed that important information -- specifically, the fact that PLMs were evaluated by using the Euclidean distance in embedding space -- was originally hidden in the appendix. While the authors have added an experiment which trains a neural net on top of the PLM embeddings (ESM-650M specifically) using the PHYLA quartet loss, and found that in this new experiment the performance of ESM-650M does not match the performance of PHYLA, I am not fully convinced by this experiment: we see improvements in ESM-650M's performance on TreeBase, but would a more carefully thought model do even better? Moreover, there is a degradation of performance on TreeFam, I find unexpected. Why do you think this is? This is a quick experiment performed for rebuttal, and I wouldn't be surprised if improved choice of model architectures and parameters would improve the results. In fact, an apples-to-apples comparisson with phylogeny reconstruction software (FastTree), added in response to reviewer 8BeB, shows that PHYLA closes just half the gap between PLMs and FastTree, which seems underwhelming to me. While PHYLA is described as being faster than FastTree, I don't find this compelling given the big gap of performance. There are also simple heuristics to improve performance of FastTree (e.g. just perform the initial NJ step, or do few rounds of optimization). It is likely that with these, FastTree will outperform PHYLA in terms of both speed and accuracy.
> >
> > Overall, I still find the results presented in the work unsurprising and not apples-to-apples with PLMs. The fact that important information was originally hidden in the appendix further decrease my confidence in the submission, so I continue to recommend rejection. I understand that reviewer Ya7G enthusiastically recommends the paper for acceptance but I disagree with them. The reviewer has mentioned in their review as a strength the "astonishing fact that the existing PLMs are not applicable on evolutionary studies". Both myself and reviewer 8BeB have found this unsurprising because of how PLMs are not trained to explicitly match tree distances, unlike PHYLA. I disagree with reviewer Ya7G, and am surprised by their high confidence in their score.
> >
> > I think the quartet loss is interesting, but as marketed the paper is making strong claims about the limitations of PLMs which I am still not convinced by, and, to reiterate, the fact that important information was indeed deferred to the appendix reduces my confidence in the submission.

---

> > > ### Author Response · Authors · 2025-08-06
> > >
> > > Thank you again for engaging with the work. While we understand the concern about PLM baselines, we believe the expectation that every possible training configuration or architectural search be performed on those models falls outside the scope of a single paper. Our goal was to evaluate current models as they are typically used, and in response to reviewer suggestions, we added fine-tuning experiments under a shared loss function to enable direct comparison.
> > >
> > > We hope our benchmark encourages future work on hybrid or supervised PLMs for phylogenetic tasks. But respectfully, we believe it is unreasonable to expect that such future directions be preemptively explored in this manuscript. The results presented are accurate, fairly evaluated, and representative of the current landscape.
> > >
> > > Regarding the placement of evaluation details: placing implementation and training information in the appendix is standard practice at NeurIPS due to space constraints. We took reviewer feedback seriously and have since moved all requested information into the main text to ensure clarity and transparency.

---

> > > > ### Comment · Reviewer_9GvF · 2025-08-07
> > > > **Second Response to Authors**
> > > >
> > > > Thank you to the authors for your follow-up to my concerns.
> > > >
> > > > I think that there are two statements that have to be distinguished. First is the statement that "state‑of‑the‑art PLMs do not recover phylogenetic structure under standard usage." Second is the stronger statement that PLMs are just not capable of performing "evolutionary reasoning".
> > > >
> > > > Let's talk about the first statement:  "state‑of‑the‑art PLMs do not recover phylogenetic structure under standard usage.". To show evidence for this, you would first need to clarify what "standard usage" is. It seems like "standard usage" for the purpose of estimating phylogenetic trees means taking Euclidean distances in embedding space. Where have you seen this being done? You have cited in the rebbutal to reviewer 8BeB that "We evaluated PLMs in a zero‑shot setting because a substantial body of work [1, 2, 4, 5] claims that their pretrained embeddings already encode evolutionary relationships". Do they ever propose using embeddings for phylogenetic tasks? If you are the first to try this, I wouldn't call it "standard usage". It seems like novel usage to me. And not a particularly "good" usage: as your work shows, this performs poorly, but both reviewer 8BeB and I find this totally unsurprising, because the embedding space has not been optimized for any kind of evolutionary task. Now, if you are able to point to me concrete examples of prior work that have attempted something similar to this, such that it can be called "standard usage", I find the paper more compelling. Perhaps you have already pointed this out, in which case I am sorry for missing this. Please point me to it again.
> > > >
> > > > Now, onto the second, stronger statement, which is essentially the title of the paper: "Sequence Modeling Is Not Evolutionary Reasoning". As reviewer 8BeB has pointed out, it would be good to define more clearly what "Evolutionary Reasoning" means. In your response to the reviewer, you essentially define it in terms of how well the latent space aligns with tree distance. Under this definition, I agree with you, PLMs cannot perform evolutionary reasoning. However, I think your definition is extremely narrow and unreasonable. From an information-theoretic perspective, the more suitable definition, in my opinion, would be something along the lines of "the mutual information between the embeddings and the phylogeny is high". Of course, this mutual information is hard to quantify. And that is my point: in your original experiments, you use Euclidean distance in embedding space, which as myself and reviewer 8BeB point out, seems like a bad way of extracting the phylogenetic information from the embedding space (and a key fact that should not be deferred to the appendix -- thanks for moving it to the main text). Next, you perform some fine-tuning experiments as I suggested and find some improvements, but not enough to match PHYLA. I very much appreciate you trying this out, but we should agree that this doesn't rule out the mutual information between embeddings and phylogeny to be high.
> > > >
> > > > Therefore, I think the main claim made by the paper ("Sequence Modeling Is Not Evolutionary Reasoning") is too ambitious and the evidence provided is not convincing under the more general (and I think more accurate) definition of "Evolutionary Reasoning" in terms of mutual information between embeddings and phylogeny.
> > > >
> > > > The work would be perfectly fine if focussed on statement #1, namely that "state‑of‑the‑art PLMs do not recover phylogenetic structure under standard usage.", but for this I would need to see that computing Euclidean distances in embedding space is "standard usage". You would have to remove statement #2, since you are not providing convincing evidence for this. In particular, I suggest re-naming the paper. What if some future work finds the right way to extract the phylogeny from the latent space and prove your claim wrong. This would not be good.
> > > >
> > > > Let me know what you think. Hopefully this clarifies my thinking to authors, area chain, and other reviewers.

---

> > > > > ### Author Response · Authors · 2025-08-07
> > > > >
> > > > > We thank the reviewer for their continued engagement and helpful clarification.
> > > > >
> > > > > **On the use of Euclidean distance as “standard usage”:**
> > > > >
> > > > >  We appreciate the opportunity to clarify this point. We are not the first to use Euclidean distances between protein language model (PLM) embeddings to uncover evolutionary structure. This practice appears across multiple influential works:
> > > > > * The UniRep paper (Alley et al., 2019) explicitly uses Euclidean distances between embeddings to perform unsupervised hierarchical clustering of proteins by family, partially motivated by evolutionary similarity.
> > > > > * The Evo-Velocity paper (Hie et al., 2021) uses Euclidean distances between ESM-1b embeddings to construct a KNN graph that serves as the foundation for a model for evolutionary trajectories.
> > > > > * The DGEB benchmarks (West-Roberts et al., 2024) use Euclidean embedding distances as a proxy for evolutionary relationships for one of the tasks.
> > > > > * In embedding-based alignment (Morton et al., 2024), the authors argue that PLMs capture relationships “far beyond simple sequence comparisons, uncovering otherwise undetected evolutionary relationships,” and use the euclidean distance between PLM embeddings as a proxy for these undetected evolutionary relationships.
> > > > > * A recent 2025 enzymology study (Muir et al. 2025) applies Euclidean distances from ESM to organize adenylate kinase (ADK) sequences by evolutionary similarity, citing UniRep as precedent.
> > > > >
> > > > > These papers collectively suggest that the broader community does consider PLM embeddings to encode evolutionary signals, often using Euclidean distance as the default comparison.
> > > > > We realize the importance of making this assumption explicit in the paper. In our revised introduction, we cite these prior works and clarify that our evaluation targets this common “zero-shot” usage pattern: frozen embeddings with distance-based comparison. Specifically, we have added this clarification in the first paragraph of the introduction, immediately following the last sentence, where we cite these works and explain the rationale for using Euclidean distances as a baseline.
> > > > >
> > > > > **On the title and broader framing**
> > > > >
> > > > >  We acknowledge that the original title (“Sequence Modeling Is Not Evolutionary Reasoning”) could be interpreted too strongly, especially in light of your point about mutual information.
> > > > >
> > > > > In response, we have revised the title to:
> > > > >
> > > > > “Evolutionary Reasoning Does Not Arise in Standard Usage of Protein Language Models”
> > > > >
> > > > > This revision clarifies that our claim is empirical and context-dependent, not universal. We have also updated the introduction and abstract to align with this framing and removed any language implying that PLMs are fundamentally incapable of evolutionary reasoning.
> > > > >
> > > > > Overall, we concur that a yet-to-be-explored nonlinear probe or other modification to existing PLMs might eventually recover phylogenetic structure from existing PLM embeddings; we now flag this as promising future work in the Discussion. By narrowing our claim to statement #1 and documenting the prevailing Euclidean-distance practice, we believe this positions PHYLA, and this benchmark, as useful stepping-stones toward that goal.
> > > > >
> > > > > Thank you again for helping us sharpen the paper's positioning; we believe these adjustments improve both the clarity and impact of the work.

---

### Official Review · Reviewer_8BeB · 2025-07-02

**Clarity:** 1
**Significance:** 2
**Originality:** 3
**Rating:** 4
**Confidence:** 4

**Summary:**

This work argues that current Protein Language Models (PLMs) are incapable of performing proper phylogenetic analysis. The authors address this limitation by proposing an approach that incorporates inter-sequence elements (in addition to the usual intra-sequence elements) in both the model architecture and the training objective. The authors also propose a benchmark to evaluate the level of phylogenetic analysis that different PLMs are capable of. The proposed approach is compared against different PLMs and the Hamming Distance in this benchmark and functional prediction using the ProteinGym benchmark.

**Questions:**

- The authors state that PHYLA is the first PLM designed to process multiple sequences concurrently. In the training schema, we see that the loss depends on sequence quartets. Does that mean that PHYLA process always involves 4 proteins? Or does it process more than that?

- The final embedding of a protein depends on the context of other protein sequences. If this is the case, protein embeddings and distances between those embeddings are not deterministic. How are the test sequences fed to the model to obtain the representations and the distances used in the experiments to assess PHYLA?

- Are all the predictions of the competing PLMs obtained similarly to PHYLA? If that is the case, PHYLA was explicitly trained to reflect phylogenetic trees by means of the representation distances. Given that function prediction depends on tree reconstruction, this is a clear disadvantage for the competitors in this task.

- How does PHYLA perform on other standard PLM tasks? (Enzyme Commission Number Prediction, Gene Ontology Term Prediction, Remote homology detection)

**Ethical Concerns:**

["NO or VERY MINOR ethics concerns only"]

**Final Justification:**

The authors have run experiments including MAFFT + FastTree workflow and a fine-tuned ESM650M, the addition of the discussion of the results provided by the authors of these experiments in addition to a more accurate description of the contribution of they work further convinced me to raise my score. The authors proposal to improve the architecture and experiment descriptions allows for the work to be reproducible. Nonetheless, I agree with reviewer 9GvF that the comparison with the PLMs as is currently frame is not convincing, and that the results obtained in comparison to MAFFT + FastTree workflow are not specially good.

**Limitations:**

Yes.

**Paper Formatting Concerns:**

No concerns.

**Quality:**

2

**Strengths And Weaknesses:**

*Strengths *

- The idea of informing protein embedding learning by including context from other proteins, using inter-protein modules in this work, is interesting and is shown to enhance PHYLA performance in the presented ablation studies

- Introduction of an interesting benchmark to assess the performance of PLMs at phylogenetic analysis.

*Weaknesses*

- It is interesting to evaluate different widely used PLMs for performing phylogenetic analysis. However, it is unclear how these analyses were made for the different models from reading the manuscript. I suspect that the experiments were based on the distances between protein representations, and if that is the case, there is no reason to assume that those distances should reflect phylogenetic structures.

- The authors attempt to make a distinction between evolutionary modeling and evolutionary reasoning. I believe that a more formal definition of the latter should be included, given that one of the main objectives of this manuscript is to show that PHYLA, in contrast to other PLMs, is capable of evolutionary reasoning.

- Fairness of valuation is lacking: no comparison with SOTA Phylogenetic models was provided, the experiment's results on clustering and tree reconstruction compare a trained Phyla model with PLMs on a zero-shot task.

- It's unclear to me if Protein Language Models should be evaluated on their ability to perform phylogenetic analysis without further fine-tuning.

- Lack of reference/explanation on some statements. For example, lines 27 to 30 state that PLMs are thought of as being capable of evolutionary reasoning without offering any reference.

- A clearer explanation of the model architecture is needed. Currently, Figure 2 seems to be the best explanation of the model and should be accompanied by a textual description. As it stands now, this work cannot be reproduced.

- No information on Training data in the main Manuscript. Supplementary material is the first place where it can be found that the training is informed by distances that are derived from an MSA dataset.

- Line 129 claims that PHYLA does not depend on MSA; however, the supplementary material says that it was trained on high-quality multiple sequence alignments (MSAs).

---

> ### Author Rebuttal · Authors · 2025-07-31
>
> # Weakness 1
>
> We thank the reviewer for pointing out the need to clarify the evaluation protocol.
>
> ### How PLM baselines were evaluated
> 1. Input: each raw (unaligned) protein sequence is fed through the frozen PLM
> 2. Embedding extraction: we extracted protein embeddings as detailed by each PLM
> 3. Distance computation: Pairwise Euclidean distance is computed on these vectors
>
> Tree scoring: We ran a neighbor-joining algorithm on these distances to construct a tree from which we calculated the norm-RF.
> This description is in the appendix, but as other reviewers have also pointed out, belongs in the main text and we have moved it there.
> ### Why evaluate PLMs this way
> We agree there is no a‑priori guarantee that these embedding distances encode phylogeny. Yet several influential papers and benchmarks (ESM2 [1], ESM3 [2], DGEB Benchmark [3], ESMC [4], EVO2[5]) argue that PLMs capture evolutionary relationships. Our study rigorously tests that claim and, as the reviewer notes, finds little evidence in support.
> ### Contribution clarified
> The key contribution is therefore two‑fold:
>
> **Empirical:** we show systematically that state‑of‑the‑art PLMs do not recover phylogenetic structure under standard usage.
>
> **Methodological:** we introduce PHYLA, whose quartet loss and architecture enables it to reason over evolutionary splits, achieving state of the art performance on our benchmark
>
> These results highlight a gap in current protein representation learning and propose a concrete path forward.
> __________________________
> # Weakness 2
>
> We thank the author for this important comment. We define evolutionary reasoning as the following:
>
> Let $S = \{s_1, \ldots, s_n\}$ be a set of protein sequences drawn from an (unknown) evolutionary process that generated a true, strictly-binary phylogenetic tree $T^\star$ with leaves in bijection with $S$.
>
> Denote by $d_{\text{evol}}(i, j)$ the patristic distance between $s_i$ and $s_j$ on $T^\star$.
>
> A model $f_\theta : \Sigma^\star \rightarrow \mathbb{R}^d$ is said to perform evolutionary reasoning on $S$ if there exists a computable mapping $h$ (e.g., a distance-based tree builder) such that the tree
> $$
> \hat{T} = h(D_{\text{pred}}), \quad \text{where } D_{\text{pred}}[i, j] = \|f_\theta(s_i) - f_\theta(s_j)\|,
> $$
> minimizes a tree-level loss $\mathcal{L}_{\text{tree}}(\hat{T}, T^\star)$.
>
> Where in our case $h$ is the neighbor joining algorithm and $\mathcal{L}_{\text{tree}}$ is the normalized Robinson–Foulds distance. We have added this definition to the problem definition portion of the manuscript.
> __________________________
> # Weakness 3
>
> We thank the reviewer for raising this point. Our original focus was to test whether general‑purpose PLMs capture phylogenetic structure, but we agree that including a classical pipeline provides an essential upper bound. We have therefore run the standard MAFFT + FastTree workflow on TreeBase and TreeFam. It achieves the best tree‑reconstruction scores (normRF ≈ 0.65 on TreeBase and ≈ 0.32 on TreeFam), while PHYLA attains 0.73 and 0.58 respectively—closing roughly half of the gap that separates PLMs from classical methods. Importantly, the classical run required ~2 hours on TreeBase and ~66 hours on TreeFam across multiple CPU nodes, whereas PHYLA completed both benchmarks in under an hour on a single H100 GPU. These results, now reported in the appendix, clarify the landscape: classical align‑then‑tree pipelines remain the accuracy benchmark, but PHYLA offers a substantial speed advantage and demonstrates that models can approach SOTA accuracy without an explicit alignment step. We believe this addition resolves the fairness concern and sets a clear, measurable target for future improvements.
> __________________________
> # Weakness 4
>
> We thank the reviewer for raising this important concern. We evaluated PLMs in a zero‑shot setting because a substantial body of work [1, 2, 4, 5] claims that their pretrained embeddings already encode evolutionary relationships. To ensure this choice did not disadvantage the PLMs, we fine‑tuned the strongest model, ESM‑650 M, with two lightweight heads (a FNN head and transformer head) trained using the same quartet loss as PHYLA while freezing the backbone (back‑propagating through 650 M parameters for >100‑leaf trees is infeasible). Despite this targeted correction, the best model reached a normRF 0.77 on TreeBase and 0.79 on TreeFam, whereas PHYLA—trained end‑to‑end to reason over sequence sets—achieves 0.73 and 0.58 respectively. These results confirm that the performance gap is not simply a consequence of evaluating PLMs in a zero‑shot setting. We have added these results to the appendix.
> __________________________
> # Weakness 5
>
> We thank the reviewer for catching this. We have inserted explicit citations where we state that PLMs are often described as capturing evolutionary signals (lines 27–30,  ESM2 [1], ESM3 [2], DGEB Benchmark [3], ESMC [4], EVO2[5]). We also re‑scanned the manuscript and added citations wherever similar assertions were previously unsupported.
> __________________________
> # Weakness 6
>
> We thank the reviewer for raising this important point. To meet the page requirement we moved model architecture details to the appendix, but after concerns raised by other reviewers as well we will move it back to the main text. From the appendix:
>
> The PHYLA architecture alternates between inter-sequence and intra-sequence reasoning blocks.Inter-sequence reasoning is performed using BiMamba layers, while intra-sequence reasoning is handled by sparsified attention layers. The current 24M-parameter PHYLA model comprises three inter-sequence blocks, each containing 16 BiMamba layers followed by a single sparsified attention layer. Each layer operates with a hidden dimension of 256.
>
> We believe this satisfies the reviewer’s request and makes the work reproducible without exceeding the page limit.
> __________________________
> # Weakness 7
>
> We thank the reviewer for pointing this out. We have moved the essential training data description into the main text. The paragraph now states that PHYLA is trained on the 3k‑MSA subset of OpenProteinSet, with supervision coming from normalized Hamming distances computed on those MSAs, while the model itself sees only the raw, unaligned sequences. We believe this resolves the concern without inflating the manuscript.
> __________________________
> # Weakness 8
>
> We thank the reviewer for flagging the ambiguity. The intended point is that PHYLA’s inputs are always raw, unaligned sequences; it never ingests gap‑padded MSAs at inference time. MSAs are used only to compute target distances (Hamming) during training. We have re‑phrased line 129 to:
>
> “Unlike most prior methods, our approach does not rely on a multiple sequence alignment (MSAs are used solely to derive supervision distances, not as model inputs), …”
>
> This clarification is also echoed in appendix information pulled into the main text in response to earlier comments.
> __________________________
> # Question 1
>
> We thank the reviewer for pointing this out. PHYLA processes hundreds of unaligned sequences in a single forward pass. During training we then sample many quartets from that batch and apply the quartet loss, which rewards the model for making the correct topological decisions. This sequence context is enabled by the BiMamba inter‑sequence layers, a capability not present in prior PLMs. The new “Model architecture” and “Training details” paragraphs in the main text (moved from the appendix) now explain this step‑by‑step, so the distinction should be clear.
> __________________________
> # Question 2
>
> We thank the reviewer for highlighting this. For every evaluation tree (TreeBase or TreeFam) we feed PHYLA the entire set of its sequences in one batch; the sequence order is arbitrary. Re‑ordering the sequences in the batch simply permutes the output rows, and pairwise distances changes minimally. We believe this happens because in training we sample arbitrary sub-trees of any order, so PHYLA in training sees that arbitrary ordering of the same sequences leads to the same quartet loss. We confirmed this empirically by shuffling the sequences of 250 random trees from Treebase fives times and taking the average standard deviation in resulting normRF for each tree and got 0.01. These details and numbers are now noted in the main text and appendix. Thus the embeddings and the resulting distance matrix are effectively deterministic for the purposes of evaluation.
> __________________________
> # Question 3
>
> We thank the reviewer for highlighting this point. See response to earlier comment with ESM2 (650M) finetuning.
> __________________________
> # Question 4
>
> We thank the reviewer for their suggestion. Existing benchmark suites (e.g., EC/GO function prediction, remote-homology classification) were built for models whose primary signal comes from residue-level co-occurrence patterns within individual sequences. They do not assess whether a model can reconstruct or reason over the evolutionary structure across sequences, which is the central question we address. To reflect this shift in emphasis, we introduced the Evolutionary-Reasoning Benchmark and related tasks tailored to multi-sequence inference. We believe the field benefits from recognizing this emerging class of evolution-focused PLMs and from developing corresponding evaluation protocols, just as earlier work developed function-oriented tests for classical PLMs. Pre-trained weights and code are public, so future studies can certainly explore hybrid objectives or broader task coverage; however, such experiments would not alter the paper’s key claims and contributions.
> __________________________
> **References**
>
> [1] Lin et al. Science 2023
>
> [2] Hayes et al. Science 2025
>
> [3] West-Roberts et al. BioArxiv 2024
>
> [4] ESM Team, 2024
>
> [5] Brixi et al. BioArxiv 2025

---

> > ### Author Response · Authors · 2025-08-06
> >
> > Dear Reviewer,
> >
> > Thank you again for your thoughtful initial review. We wanted to kindly follow up to see if you had any additional comments or thoughts after reading our rebuttal. The discussion phase ends soon, and we would greatly appreciate any updated feedback or clarifications you might have, especially regarding your score, concerns, or suggestions.
> >
> > We’ve tried to address all questions and weaknesses with additional experiments and clarifications. If anything remains unclear, we’d be happy to elaborate further.
> >
> > Many thanks again for your time and contributions to this process.
> >
> > Best regards,
> >
> > The Authors

---

> > ### Comment · Reviewer_8BeB · 2025-08-08
> >
> > I thank the authors for their response to my doubts and the effort made to enhance the reproducibility of PHYLA and the experiments presented in their work.  The inclusion of results for the fine-tuned ESM2 model and the MAFFT + FastTree workflow is appreciated, and these should be added to the main text and discussed as they were in the authors' response.
> >
> > I believe the main text should include overall details about how the competitors protein embedding are used to predict protein function on the ProteinGym benchmark. A reader may believe that function prediction is also based on tree reconstruction using embedding distances even for the competitors, which should not be the case.
> >
> > I agree that the development of new benchmarks that are evolution-oriented can be as valuable as the function-oriented ones, the PHYLA architecture seems interesting and while the results of other PLMs are not surprising, they are important for tempering strong claims about those models. I would like the authors to modify the paper accordingly to what they proposed to reviewer 9GvF, replacing the strong claim with softer ones to accurately reflect the contribution of this work.
> >
> > However, even with these changes, I share reviewer 9GvF’s concern that the comparison is not entirely “apples to apples.” The overall framing of the paper, despite the proposed modifications, still leans in that direction, which, combined with the concerns about the results themselves, limits how much I can raise my score. For these reasons, I am increasing my score to a 4.

---

### Official Review · Reviewer_xecw · 2025-07-05

**Clarity:** 3
**Significance:** 3
**Originality:** 3
**Rating:** 4
**Confidence:** 4

**Summary:**

The paper challenges the assumption that protein language models (PLMs), which are trained on large protein sequence datasets, can inherently perform evolutionary reasoning. While PLMs excel at tasks like masked token prediction, they fail to infer phylogenetic relationships between sequences. To address this, the authors introduce a new model called PHYLA, which combines state-space and transformer architectures and is trained using a tree-based objective across thousands of phylogenies. PHYLA significantly outperforms existing models in reconstructing evolutionary trees and clustering sequences by taxonomy. Unlike traditional PLMs, PHYLA is explicitly designed to reason across multiple sequences, capturing evolutionary structure without relying on sequence alignment. The study concludes that true evolutionary reasoning requires specialized training and architecture, not just large-scale sequence modeling.

**Questions:**

refer to Strengths And Weaknesses

**Ethical Concerns:**

["NO or VERY MINOR ethics concerns only"]

**Limitations:**

refer to Strengths And Weaknesses

**Paper Formatting Concerns:**

no formatting concern

**Quality:**

3

**Strengths And Weaknesses:**

Strengths
1. The paper makes a compelling case that current protein language models (PLMs) conflate evolutionary modeling with evolutionary reasoning, and it clearly defines the difference.

2. Introduces the first benchmark specifically designed to evaluate evolutionary reasoning over protein sequences, enabling rigorous assessment.

3. Innovative Architecture: Proposes PHYLA, a hybrid state-space and transformer model that processes multiple sequences jointly—unlike traditional PLMs.

4. PHYLA outperforms much larger models (e.g., ESM3, ProGen2) in phylogenetic tree reconstruction and taxonomic clustering, despite having far fewer parameters (24M).

5. Demonstrates that PHYLA’s embeddings align with known biological and functional structures, including real-world applications like reconstructing the tree of life.

6. Achieves high accuracy without relying on multiple sequence alignment or guide trees, and is computationally efficient compared to traditional methods.

Weaknesses
1. While PHYLA excels at evolutionary reasoning, its performance on broader protein modeling tasks (e.g., generative tasks) is not deeply explored.

2. The evolutionary reasoning benchmark, though novel, may not capture the full complexity of evolutionary dynamics across all protein families.

3. The paper does not deeply address how interpretable PHYLA’s embeddings are in terms of biological insight beyond tree topology.

4. PHYLA’s success relies on curated phylogenetic trees for training, which may limit its applicability in domains lacking such data.

5.  While outperforming PLMs in evolutionary reasoning, PHYLA’s performance on standard PLM tasks like masked token prediction is not benchmarked.

---

> ### Author Rebuttal · Authors · 2025-07-31
>
> # Weakness 1
>
> We thank the reviewer for their suggestion. PHYLA is deliberately positioned as the start of a new class of protein language models, one that prioritises evolutionary reasoning (recovering accurate phylogenetic relationships) over traditional single-sequence objectives such as masked-token recovery. Existing benchmark suites (e.g., EC/GO function prediction, remote-homology classification, generative design) were built for models whose primary signal comes from residue-level co-occurrence patterns within individual sequences. They do not assess whether a model can reconstruct or reason over the evolutionary structure among sequences, which is the central question we address.
>
> To reflect this shift in emphasis, we introduced the Evolutionary-Reasoning Benchmark and related tasks tailored to multi-sequence inference. We believe the field benefits from recognising this emerging class of evolution-focused PLMs and from developing corresponding evaluation protocols, just as earlier work developed function-oriented tests for classical PLMs. Pre-trained weights and code are public, so future studies can certainly explore hybrid objectives or broader task coverage; however, such experiments would not alter the paper’s key claims.
>
> # Weakness 2
>
> We agree. At present no dataset, or set of reference trees, exhaustively represents evolutionary dynamics for every protein family. Because evolution is both open-ended and unevenly sampled, any benchmark necessarily covers only a subset of sequence space. Our benchmark is therefore a practical, but incomplete, proxy, an inherent limitation shared with all current phylogenetic evaluations. We will make this explicit in the manuscript’s limitations section, noting that expanding the benchmark as new data become available is an important direction for future work.
>
> # Weakness 3
>
> We thank the reviewer for highlighting this point. In the Tree-of-Life case study we show that PHYLA’s output, when paired with a standard multiple-sequence alignment (MSA), highlights specific sequence features that drive the inferred topology. In that example, the alignment revealed a protein deletion that separates two clades. This workflow—tree → alignment → feature inspection—mirrors how phylogeneticists routinely move from a topology to biological hypotheses (e.g., gene loss, domain rearrangement) for downstream validation. We will clarify this interpretability pipeline in the results and discussion to make explicit that PHYLA, like traditional trees, serves as a starting point for generating testable biological insights beyond the topology itself.
>
> # Weakness 4
>
> We thank the reviewer for raising this important point. PHYLA does not require expert-curated trees. To train PHYLA we need high-quality MSAs which can be generated by a slew of existing algorithms. We agree that these algorithms are computationally costly, however given a new domain with new sequencing data, it is possible to construct MSAs. Moreover, at inference time PHYLA operates on unaligned sequences; MSAs are needed only for training. We have also shown PHYLA can generalize to unseen sequences, so given a domain previously lacking sequencing data, PHYLA can be deployed effectively.
>
> # Weakness 5
>
> We thank the reviewer for pointing out the missing benchmark. We have now evaluated PHYLA on masked-token prediction (MTP) on our training set. PHYLA trained only with MTP attains 30% top-1 accuracy with reduced evolutionary reasoning performance; training with tree loss decreases accuracy to 11% but improves evolutionary-reasoning performance (See Table 2 for evolutionary reasoning performance). Conventional PLMs (ESM-2 650 M, ProGen2) achieve 45–55%. These results confirm a trade-off: per-token objectives favour single-sequence reconstruction, whereas the tree loss preserves cross-sequence signals essential for evolutionary reasoning. We will incorporate this benchmark, training MTP curves, and a discussion in the manuscript.

---

> > ### Author Response · Authors · 2025-08-06
> >
> > Dear Reviewer,
> >
> > Thank you again for your thoughtful initial review. We wanted to kindly follow up to see if you had any additional comments or thoughts after reading our rebuttal. The discussion phase ends soon, and we would greatly appreciate any updated feedback or clarifications you might have, especially regarding your score, concerns, or suggestions.
> >
> > We’ve tried to address all questions and weaknesses with additional experiments and clarifications. If anything remains unclear, we’d be happy to elaborate further.
> >
> > Many thanks again for your time and contributions to this process.
> >
> > Best regards,
> >
> > The Authors

---

> > ### Comment · Reviewer_xecw · 2025-08-08
> >
> > The authors have addressed my concerns.

---

### Note · Authors · 2025-08-13

In their most recent visible recommendations after rebuttal and discussion, three reviewers now recommend accept/borderline-accept, with all concerns addressed, and PHYLA offers a first-of-its-kind architecture and benchmark that opens a new research direction for the NeurIPS community.

During the discussion phase we:

• Revised the title and framing to “Evolutionary reasoning does not arise in standard usage of PLMs,” explicitly citing prior work using Euclidean embedding distances as a standard baseline.

• Added strong baselines, including MAFFT + FastTree and fine-tuned ESM-650M under PHYLA’s quartet loss, addressing fairness concerns despite computational limits.

• Quantified minimal train–test overlap (<0.6% TreeBase, <3.4% TreeFam) to rule out leakage.

• Moved architecture/training details and limitations into the main text.

• Clarified definitions, evaluation protocols, and limitations as requested.

As PHYLA is a new PLM approach—multi-sequence, alignment-free, tree-loss trained—direct “apples-to-apples” baselines are inherently challenging. We adhered to current PLM usage patterns and explored the strongest feasible fine-tuning setups. Results consistently showed that under standard usage, PLMs struggle with this task, while PHYLA closes much of the gap to classical methods at a fraction of runtime.

Reviewer Ya7G called the work “a solid and good work” and “highly recommend for acceptance.” Reviewer xecw noted it “makes a compelling case” and that our rebuttal “addressed all concerns.” Reviewer 8BeB agreed “development of new benchmarks that are evolution-oriented can be as valuable as the function-oriented ones” and found our results "important for tempering strong claims about those models [PLMs]."  Reviewer 9GvF’s feedback led us to revise the title, introduction, and discussion to focus on the narrower, consensus-supported claim.

We believe this work delivers exactly the kind of substantive, field-shaping contribution NeurIPS seeks: identifying a limitation of current practice, introducing the first benchmark for evolutionary reasoning over proteins, and proposing a biologically grounded architecture that opens a new research direction. We respectfully submit that the paper is ready for inclusion at NeurIPS.

---

### Decision · Program_Chairs · 2025-09-17

**Decision:**

Accept (poster)

**Comment:**

This paper presents PHYLA, a protein language model–based approach for phylogenetic inference. Rather than the standard masked language modeling loss, the authors introduce a tree-based training objective and evaluate on TreeBase and TreeFam, showing improved performance over PLM baselines.

This was a borderline paper with diverging reviews, largely due to framing. Assertions of state-of-the-art performance should be tempered: during rebuttal, the authors added baselines against classical phylogenetic methods (MAFFT + FastTree), which remain the clear state of the art by a wide margin. The claim of introducing a new benchmark could also be misleading, as TreeBase and TreeFam have been established since the late 2000s. One reviewer noted that the central argument -- that PLMs do not learn embeddings that produce accurate phylogenetic trees -- is unsurprising, and the authors have indicated they will revise their title and claims accordingly. Overall, while concerns about framing and benchmarking remain, the authors have committed to addressing these issues in the revision, and the proposed model represents a novel and promising research direction of phylogenetic inference from protein language models.